# Model-Based Episodic Memory Induces Dynamic Hybrid Controls

**Hung Le, Thommen Karimpanal George, Majid Abdolshah, Truyen Tran, Svetha Venkatesh**
Applied AI Institute, Deakin University, Geelong, Australia
`thai.le@deakin.edu.au`

## Abstract

Episodic control enables sample efficiency in reinforcement learning by recalling past experiences from an episodic memory. We propose a new model-based episodic memory of trajectories addressing current limitations of episodic control. Our memory estimates trajectory values, guiding the agent towards good policies. Built upon the memory, we construct a complementary learning model via a dynamic hybrid control unifying model-based, episodic and habitual learning into a single architecture. Experiments demonstrate that our model allows significantly faster and better learning than other strong reinforcement learning agents across a variety of environments including stochastic and non-Markovian settings.

## 1 Introduction

Episodic memory or "mental time travel" [6] allows recreation of past experiences. In reinforcement learning (RL), episodic control (EC) uses this memory to control behavior, and complements forward model and simpler, habitual (cached) control methods. The use of episodic memory[1] is shown to be very useful in early stages of RL [29, 4] and backed up by cognitive evidence [42, 41]. Using only one or few instances of past experiences to make decisions, EC agents avoid complicated planning computations, exploiting experiences faster than the other two control methods. In hybrid control systems, EC demonstrates excellent performance and better sample efficiency [36, 30].

Early works on episodic control use tabular episodic memory storing a raw trajectory as a sequence of states, actions and rewards over consecutive time steps. To select a policy, the methods iterate through all stored sequences and are thus only suitable for small-scale problems [29, 9]. Other episodic memories store individual state-action pairs, acting as the state-action value table in tabular RL, and can generalize to novel states using nearest neighbor approximations [4, 36]. Recent works [35, 16, 30, 46] leverage both episodic and habitual learning by combining state-action episodic memories with Q-learning augmented with parametric value functions like Deep Q-Network (DQN; [34]). The combination of the "fast" non-parametric episodic and "slow" parametric value facilitates Complementary Learning Systems (CLS) – a theory posits that the brain relies on both slow learning of distributed representations (neocortex) and fast learning of pattern-separated representations (hippocampus) [31].

Existing episodic RL methods suffer from 3 issues: (a) near-deterministic assumption [4] which is vulnerable to noisy, stochastic or partially observable environments causing ambiguous observations; (b) sample-inefficiency due to storing state-action-value which demands experiencing all actions to make reliable decisions and inadequate memory writings that prevent fast and accurate value propagation inside the memory [4, 36]; and finally, (c) assuming fixed combination between episodic and parametric values [30, 16] that makes episodic contribution weight unchanged for different observations and requires manual tuning of the weight. We tackle these open issues by designing

---

[1]The episodic memory in this setting is an across-lifetime memory, persisting throughout training.

35th Conference on Neural Information Processing Systems (NeurIPS 2021).

a novel model that flexibly integrates habitual, model-based and episodic control into a single architecture for RL.

To tackle issue (a) the model learns representations of the trajectory by minimizing a self-supervised loss. The loss encourages reconstruction of past observations, thus enforcing a compressive and noise-tolerant representation of the trajectory for the episodic memory. Unlike model-based RL [39, 13] that simulates the world, our model merely captures the trajectories.

To address issue (b), we propose a model-based value estimation mechanism established on the trajectory representations. This allows us to design a memory-based planning algorithm, namely Model-based Episodic Control (MBEC), to compute the action value online at the time of making decisions. Hence, our memory does not need to store actions. Instead, the memory stores trajectory vectors as the keys, each of which is tied to a value, facilitating nearest neighbor memory lookups to retrieve the value of an arbitrary trajectory (memory read). To hasten value propagation and reduce noise inside the memory, we propose using a weighted averaging write operator that writes to multiple memory slots, plus a bootstrapped refine operator to update the written values at any step.

Finally, to address issue (c), we create a flexible CLS architecture, merging complementary systems of learning and memory. An episodic value is combined with a parametric value via dynamic consolidation. Concretely, conditioned on the current observation, a neural network dynamically assigns the combination weight determining how much the episodic memory contributes to the final action value. We choose DQN as the parametric value function and train it to minimize the temporal difference (TD) error (habitual control). The learning of DQN takes episodic values into consideration, facilitating a distillation of the episodic memory into the DQN's weights.

Our contributions are: (i) a new model-based control using episodic memory of trajectories; (ii) a Complementary Learning Systems architecture that addresses limitations of current episodic RL through a dynamic hybrid control unifying model-based, episodic and habitual learning (see Fig. 1); and, (iii) demonstration of our architecture on a diverse test-suite of RL problems from grid-world, classical control to Atari games and 3D navigation tasks. We show that the MBEC is noise-tolerant, robust in dynamic grid-world environments. In classical control, we show the advantages of the hybrid control when the environment is stochastic, and illustrate how each component plays a crucial role. For high-dimensional problems, our model also achieves superior performance. Further, we interpret model behavior and provide analytical studies to validate our empirical results.

## 2 Background

### 2.1 Deep Reinforcement Learning

Reinforcement learning aims to find the policy that maximizes the future cumulative rewards of sequential decision-making problems [40]. Model-based approaches build a model of how the environment operates, from which the optimal policy is found through planning [39]. Recent model-based RL methods can simulate complex environments enabling sample-efficiency through allowing agents to learn within the simulated "worlds" [14, 23, 15]. Unlike these works, Q-learning [43] – a typical model-free method, directly estimates the true state-action value function. The function is defined as $Q(s, a) = \mathbb{E}_\pi \left[ \sum_t \gamma^t r_t \mid s, a \right]$, where $r_t$ is the reward at timestep $t$ that the agent receives from the current state $s$ by taking action $a$, followed policy $\pi$. $\gamma \in (0, 1]$ is the discount factor that weights the importance of upcoming rewards. Upon learning the function, the best action can be found as $a_t = \underset{a}{\mathrm{argmax}} \, Q(s_t, a)$.

With the rise of deep learning, neural networks have been widely used to improve reinforcement learning. Deep Q-Network (DQN; [34]) learns the value function $Q_\theta(s, a)$ using convolutional and feed-forward neural networks whose parameters are $\theta$. The value network takes an image representation of the state $s_t$ and outputs a vector containing the value of each action $a_t$. To train the networks, DQN samples observed transition $(s_t, a_t, r_t, s_{t+1})$ from a replay buffer to minimize the squared error between the value output and target $y_t = r_t + \gamma \underset{a}{\max} \, Q'_\theta(s_{t+1}, a)$ where $Q'_\theta$ is the target network. The parameter of the target network is periodically set to that of the value network, ensuring stable learning. The value network of DQN resembles a semantic memory that gradually encodes the value of state-action pairs via replaying as a memory consolidation in CLS theory [24].

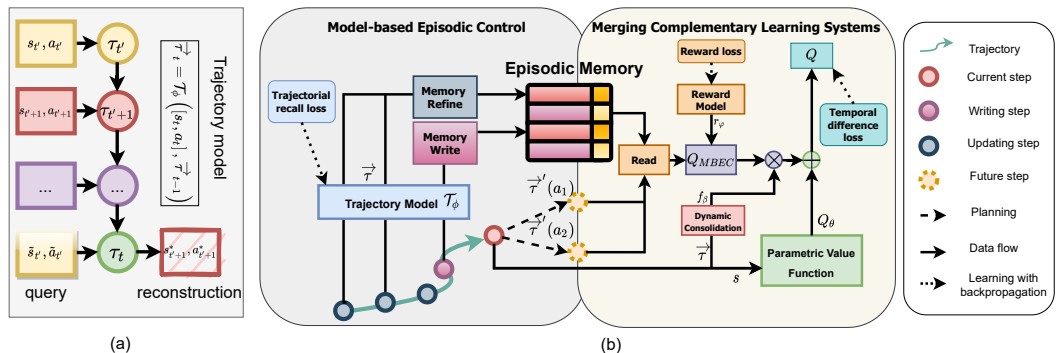

Figure 1: (a) Trajectorial Recall. The trajectory model reconstructs any past observation along the trajectory given noisy preceding s-a pair as a cue. (b) Dynamic hybrid control with the episodic memory at its core. The trajectory model, trained with TR loss (Eq. 1), encodes representations for writing (Eq. 2) and updating (Eq. 4) the episodic memory. Model-based Episodic Control (MBEC) plans actions (e.g. $a_1$ and $a_2$) and computes future trajectory representations ($\overrightarrow{\tau}'(a_1)$ and $\overrightarrow{\tau}'(a_2)$) for reading the memory's stored values. The read-out, together with the reward model, estimates the episodic value $Q_{MBEC}$ (Eq. 6). The Complementary Learning Systems (CLS) combines $Q_{MBEC}$ and the traditional semantic $Q_\theta$ using dynamic consolidation conditioned on $\overrightarrow{\tau}$ (Eq. 7). The parameters of the CLS are optimized with TD loss (Eq. 8; habitual control).

Experience replay is critical for DQN, yet it is slow, requiring a lot of observations since the replay buffer only stores raw individual experiences. Prioritized Replay [37] improves replaying process with non-uniform sampling favoring important transitions. Others overcome the limitation of one-step transition by involving multi-step return in calculating the value [28, 18]. These works require raw trajectories and parallel those using episodic memory that persists across episodes.

## 2.2 Memory-based Controls

Episodic control enables sample-efficiency through explicitly storing the association between returns and state-action pairs in episodic memory [29, 4, 36]. When combined with Q-learning (habitual control), the episodic memory augments the value function with episodic value estimation, which is shown beneficial to guide the RL agent to latch on good policies during early training [30, 46, 20].

In addition to episodic memory, Long Short-Term Memory (LSTM; [19]) and Memory-Augmented Neural Networks (MANNs; [10, 11]) are other forms of memories that are excel at learning long sequences, and thus extend the capability of neural networks in RL. In particular, Deep Recurrent Q-Network (DRQN; [17]) replaces the feed-forward value network with LSTM counterparts, aiming to solve Partially-Observable Markov Decision Process (POMDP). Policy gradient agents are commonly equipped with memories [33, 11, 38]. They capture long-term dependencies across states, enrich state representation and contribute to making decisions that require reference to past events in the same episode.

Recent works use memory for reward shaping either via contribution analysis [1] or memory attention [21]. To improve the representation stored in memory, some also construct a model of transitions using unsupervised learning [44, 13, 8]. As these memories are cleared at the end of the episode, they act more like working memory with a limited lifespan [2]. Relational [45, 26] and Program Memory [27] are other forms of memories that have been used for RL. They are all different from the notion of persistent episodic memory, which is the main focus of this paper.

## 3 Methods

We introduce a novel model that combines habitual, model based and episodic control wherein episodic memory plays a central role. Thanks to the memory storing trajectory representations, we can estimate the value in a model-driven fashion: for any action considered, the future trajectory is computed to query the episodic memory and get the action value. This takes advantage of model-based planning and episodic inference. The episodic value is then fused with a slower, parametric value function to leverage the capability of both episodic and habitual learning. Fig. 1 illustrates

**Algorithm 1** MBEC++: Complementary reinforcement learning with MBEC and DQN.

1: $\mathcal{D}$: Replay buffer; $\mathcal{B}$: Trajectory buffer; $\mathcal{C}$: Chunk buffer; $\mathcal{M}$: Episodic memory; $\mathcal{T}_\phi$: Trajectory model; $r_\varphi(s, a)$: Reward model; $L$: Chunk length
2: **for each** episode **do**
3:     Initialize $\overrightarrow{\tau}_0 = 0$; $\mathcal{B} = \emptyset$; $\mathcal{C} = \emptyset$
4:     **for** $t = 1, T$ **do**
5:         Observe state $s_t$. Compute $f_\beta(\overrightarrow{\tau}_{t-1})$.
6:         Select action $a_t \leftarrow \epsilon$-greedy policy using Q in Eq. 7
7:         Observe reward $r_t$. Move to next state $s_{t+1}$. Compute $\overrightarrow{\tau}_t \leftarrow \mathcal{T}_\phi([s_t, a_t], \overrightarrow{\tau}_{t-1})$
8:         Add $(s_t, a_t, s_{t+1}, r_t, \overrightarrow{\tau}_{t-1}, \overrightarrow{\tau}_t)$ and $(s_t, a_t)$ to $\mathcal{D}$ and $\mathcal{B}$, respectively.
9:         Refine memory: $\mathcal{M} \leftarrow \text{refine}(s_t, \overrightarrow{\tau}_{t-1} | \mathcal{M})$
10:         **for** $(s_{t'}, a_{t'}, s_{t'+1}, r_{t'}, \overrightarrow{\tau}_{t'-1}, \overrightarrow{\tau}_{t'})$ sampled from $\mathcal{D}$ **do**
11:             Compute $Q(s, a)$ using $s_{t'}, a_{t'}, \overrightarrow{\tau}_{t'-1}$ (Eq. 7)
12:             Compute $Q(s', a')$ using $s_{t'+1}, \overrightarrow{\tau}_{t'} \forall a'$ (Eq. 7)
13:             Optimize $\mathcal{L}_q$ wrt $\theta$ and $\beta$ (Eq. 8)
14:             Compute $\mathcal{L}_{re}$ using $s_{t'}, a_{t'}, r_{t'}$ (Eq. 5). Optimize $\mathcal{L}_{re}$ wrt $\varphi$
15:         **if** $t \mod L == 0$ **then**
16:             Add $(\overrightarrow{\tau}_{t-1}, r_t)$ to $\mathcal{C}$. Sample $(s_{t'}, a_{t'})$ from $\mathcal{B}$ and optimize $\mathcal{L}_{tr}$ wrt $\phi$ and $\omega$ (Eq. 1)
17:         **if** $t == T$ **then**
18:             **for each** $\overrightarrow{\tau}_i \in \mathcal{C}$ **do**
19:                 Compute $\hat{V}(\overrightarrow{\tau}_i) = \sum_{j=i}^{T-1} \gamma^{j-i} r_{j+1}$. Write $\mathcal{M} \leftarrow \text{write}(\overrightarrow{\tau}_i, \hat{V}(\overrightarrow{\tau}_i) | \mathcal{M})$

these components. We first present the formation of the trajectory representation and the episodic memory. Next, we describe how we estimate the value from this memory and parametric Q networks.

### 3.1 Episodic Memory Stores Trajectory Representations

In this paper, a trajectory $\tau_t$ is a sequence of what happens up to the time step $t$: $\tau_t = [(s_1, a_1), ..., (s_t, a_t)]$. If we consider $\tau_t$ as a piece of memory that encodes events in an episode, from that memory, we must be able to recall any past event. This ability in humans can be examined through the serial recall test wherein a previous item cues to the recall of the next item in a sequence [7]. We aim to represent trajectories as vectors manifesting that property. As such, we employ a recurrent trajectory network $\mathcal{T}_\phi$ to model $\overrightarrow{\tau}_t = \mathcal{T}_\phi([s_t, a_t], \overrightarrow{\tau}_{t-1})$ where $\mathcal{T}_\phi$ is implemented as an LSTM [19] and $\overrightarrow{\tau}_t \in \mathbb{R}^H$ is the vector representation of $\tau_t$ and also the hidden state of the LSTM.

We train the *trajectory model* $\mathcal{T}_\phi$ to compute $\overrightarrow{\tau}_t$ such that it is able to reconstruct the *next* observation of any *past* experience, simulating the serial recall test. Concretely, given a noisy version $(\tilde{s}_{t'}, \tilde{a}_{t'})$ of a query state-action $(s_{t'}, a_{t'})$ sampled from a trajectory buffer $\mathcal{B}$ at time step $t' < t$, we minimize the *trajectorial recall (TR) loss* as follows,

$$\mathcal{L}_{tr} = \mathbb{E}\left(\|y^*(t) - [s_{t'+1}, a_{t'+1}]\|_2^2\right) \tag{1}$$

where $y^*(t) = \mathcal{G}_\omega(\mathcal{T}_\phi([\tilde{s}_{t'}, \tilde{a}_{t'}], \overrightarrow{\tau}_t))$, $\mathcal{G}_\omega$ is a reconstruction function, implemented as a feedforward neural network. The trajectory network $\mathcal{T}_\phi$ must implement some form of associative memory, compressing information of any state-action query in the past in its current representation $\overrightarrow{\tau}_t$ to reconstruct the next observation of the query, keeping the TR loss low (see a visualization in Fig. 1 (a)). Appendix A.1 theoretically explains why the TR loss is suitable for episodic control.

Our goal is to keep the trajectory representation and its associated value as the key and value of an episodic memory $\mathcal{M} = \{\mathcal{M}^k, \mathcal{M}^v\}$, respectively. $\mathcal{M}^k \in \mathbb{R}^{N \times H}$ and $\mathcal{M}^v \in \mathbb{R}^{N \times 1}$, where $N$ is the maximum number of memory slots. The true value of a trajectory $\overrightarrow{\tau}_t$ is simply defined as the value of the resulting state of the trajectory $V(\overrightarrow{\tau}_t) = V(s_{t+1}) = \mathbb{E}\left(\sum_{i=0}^{T-t-1} \gamma^i r_{t+1+i}\right)$, $T$ is the terminal step. The memory stores estimation of the true trajectory values through averaging empirical returns by our weighted average writing mechanism (see in the next section).

## 3.2 Memory Operators

**Memory reading** Given a query key $\overrightarrow{\tau}$, we read from the memory the corresponding value by referring to neighboring representations. Concretely, two reading rules are employed

$$\text{read}\left(\overrightarrow{\tau}|\mathcal{M}\right) = \begin{cases} \sum_{i\in\mathcal{N}^{K_r}(\overrightarrow{\tau})} \frac{\langle\mathcal{M}_i^k,\overrightarrow{\tau}\rangle\mathcal{M}_i^v}{\sum_{j\in\mathcal{N}^K(\overrightarrow{\tau})}\langle\mathcal{M}_j^k,\overrightarrow{\tau}\rangle} & (a) \\ \max_{i\in\mathcal{N}^{K_r}(\overrightarrow{\tau})} \mathcal{M}_i^v & (b) \end{cases}$$

where $\langle\cdot\rangle$ is a kernel function and $\mathcal{N}^{K_r}(\cdot)$ retrieves top $K_r$ nearest neighbors of the query in $\mathcal{M}^k$. The read-out is an estimation of the value of the trajectory $\overrightarrow{\tau}$ wherein the weighted average rule $(a)$ is a balanced estimation, while the max rule $(b)$ is optimistic, encouraging exploitation of the best local experience. In this paper, the two reading rules are simply selected randomly with a probability of $p_{read}$ and $1 - p_{read}$, respectively.

**Memory writing** Given the writing key $\overrightarrow{\tau}$ and its estimated value $\hat{V}(\overrightarrow{\tau})$, the write operator $\text{write}\left(\overrightarrow{\tau}, \hat{V}(\overrightarrow{\tau})|\mathcal{M}\right)$ consists of several steps. First, we add the value to the memories $\mathcal{M}^v$ if the key cannot be found in the key memory $\mathcal{M}^k$ (this happens frequently since key match is rare). Then, we update the values of the key neighbors such that the updated values are approaching the written value $\hat{V}(\overrightarrow{\tau})$ with speeds relative to the distances as $\forall i \in \mathcal{N}^{K_w}(\overrightarrow{\tau})$ :

$$\mathcal{M}_i^v \leftarrow \mathcal{M}_i^v + \alpha_w\left(\hat{V}(\overrightarrow{\tau}) - \mathcal{M}_i^v\right)\frac{\langle\mathcal{M}_i^k,\overrightarrow{\tau}\rangle}{\sum_{j\in\mathcal{N}^{K_w}(\overrightarrow{\tau})}\langle\mathcal{M}_j^k,\overrightarrow{\tau}\rangle} \tag{2}$$

where $\alpha_w$ is the writing rate. Finally, the key can be added to the key memory. When it exceeds memory capacity $N$, the earliest added will be removed. For simplicity, $K_w = K_r = K$.

We note that our memory writing allows multiple updates to multiple neighbor slots, which is unlike the single-slot update rule [4, 36, 30]. Here, the written value is the Monte Carlo return collected from $t + 1$ to the end of the episode $\hat{V}(\overrightarrow{\tau}_t) = \sum_{i=0}^{T-t-1}\gamma^i r_{t+1+i}$. Following [25], we choose to write the trajectory representation of every $L$-th time-step (rather than every time-step) to save computation while still maintaining good memorization. Appendix A.2 provides a mean convergence analysis of our writing mechanism.

**Memory refining** As the memory writing is only executed after the episode ends, it delays the value propagation inside the memory. Hence, we design the $\text{refine}\left(s_t, \overrightarrow{\tau}_{t-1}|\mathcal{M}\right)$ operator that tries to minimize the one-step TD error of the memory's value estimation. As such, at an arbitrary timestep $t$, we estimate the future trajectory if the agent takes action $a$ using the trajectory model as $\overrightarrow{\tau}'_t(a) = \mathcal{T}_\phi([s_t, a], \overrightarrow{\tau}_{t-1})$. Then, we can update the memory values as follows,

$$Q' = \max_a r_\varphi(s_t, a) + \gamma\text{read}\left(\overrightarrow{\tau}'_t(a)|\mathcal{M}\right) \quad (3) \qquad\qquad \mathcal{M} \leftarrow \text{write}\left(\overrightarrow{\tau}_{t-1}, Q'|\mathcal{M}\right) \quad (4)$$

where $r_\varphi$ is a reward model using a feed-forward neural network. $r_\varphi$ is trained by minimizing

$$\mathcal{L}_{re} = \mathbb{E}\left(r - r_\varphi(s, a)\right)^2 \tag{5}$$

The memory refining process can be shown to converge in finite MDP environments.

**Proposition 1.** *In a finite MDP $(\mathcal{S},\mathcal{A},\mathcal{T},\mathcal{R})$, given a fixed bounded $r_\varphi$ and an episodic memory $\mathcal{M}$ with* read *(average rule) and* write *operations, the memory* refine *given by Eq. 4 converges to a fixed point with probability 1 as long as $\gamma < 1$, $\sum_{t=1}^{\infty}\alpha_{w,t} = \infty$ and $\sum_{t=1}^{\infty}\alpha_{w,t}^2 < \infty$.*

*Proof.* See Appendix A.3. $\qquad\qquad\qquad\qquad\qquad\qquad\qquad\qquad\qquad\qquad\qquad\qquad\qquad\qquad\qquad$ □

## 3.3 Model-based Episodic Control (MBEC)

Our agent relies on the memory at every timestep to choose its action for the current state $s_t$. To this end, the agent first plans some action and uses $\mathcal{T}_\phi$ to estimate the future trajectory. After that,

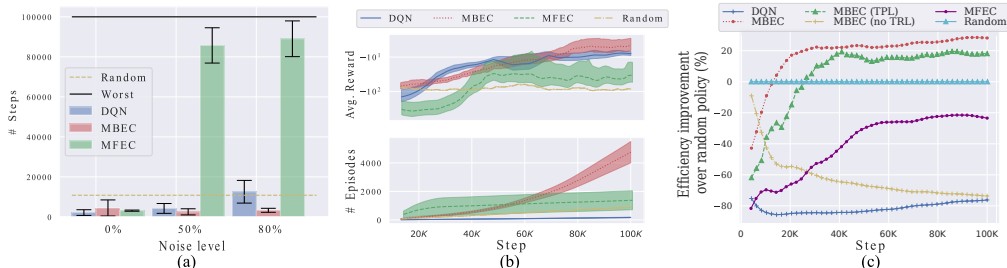

Figure 2: Maze navigation. (a) Noisy mode: number of steps required to complete 100 episodes with different noise rates (lower is better). (b) Trap mode: average reward (upper) and number of completed episodes (lower) over timesteps (higher is better). (c) Dynamic mode: efficiency improvement over random policy across timesteps (higher is better).

it reads the memory to get the value of the planned trajectory. This mechanism takes advantage of model-based RL's planning and episodic control's non-parametric inference, yielding a novel hybrid control named Model-based Episodic Control (MBEC). The state-action value then is

$$Q_{MBEC}(s, a) = r_{\varphi}(s, a) + \gamma \text{read}\left(\overrightarrow{\tau}'(a) | \mathcal{M}\right) \tag{6}$$

The MBEC policy then is $\pi(s) = \underset{a}{\text{argmax}} \, Q_{MBEC}(s, a)$. Unlike model-free episodic control, we compute on-the-fly instead of storing the state-action value. Hence, the memory does not need to store all actions to get reliable action values.

## 3.4 Model-based Episodic Control Facilitates Complementary Learning Systems

The episodic value provides direct yet biased estimation from experiences. To compensate for that, we can use a neural network $Q_{\theta}(s_t, a_t)$ to give an unbiased value estimation [34], representing the slow-learning semantic memory that gradually encodes optimal values. Prior works combine by a weighted summation of the episodic and semantic value wherein the weight is fixed [30, 16]. We believe that depending on the specific observations, we may need different weights to combine the two values. Hence, we propose to combine the two episodic and semantic systems as

$$Q(s_t, a_t) = Q_{MBEC}(s_t, a_t) f_{\beta}\left(\overrightarrow{\tau}_{t-1}\right) + Q_{\theta}(s_t, a_t) \tag{7}$$

where $f_{\beta}$ is a feed-forward neural network with sigmoid activation that takes the previous trajectory as the input and outputs a consolidating weight for the episodic value integration. This allows dynamic integration conditioned on the trajectory status. The semantic system learns to take episodic estimation into account in making decisions via replaying to minimize one-step TD error,

$$\mathcal{L}_q = \mathbb{E}\left(r + \gamma \max_{a'} Q(s', a') - Q(s, a)\right)^2 \tag{8}$$

Here we note that $Q_{MBEC}$ is also embedded in the target, providing better target value estimation in the early phase of learning when the parametric model does not learn well. We follow [34] using a replay buffer $\mathcal{D}$ to store $(s, a, s', r)$ across episodes. Without episodic contribution, TD or habitual learning is slow [16, 30]. Our episodic integration allows the agent to rely on MBEC whenever it needs to compensate for immature parametric systems. Alg. 1 (MBEC++) summarizes MBEC operations within the complementary learning system. The two components (MBEC and CLS) are linked by the episodic memory as illustrated in Fig. 1 (b).

## 4 Results

In this section, we examine our proposed episodic control both as a stand-alone (MBEC) and within a complementary learning system (MBEC++). To keep our trajectory model simple, for problems with image state, it learns to reconstruct the feature vector of the image input, rather than the image itself. The main baselines are DQN [34] and recent (hybrid) episodic control methods. Details of the baseline configurations and hyper-parameter tuning for each tasks can be found in Appendix B.

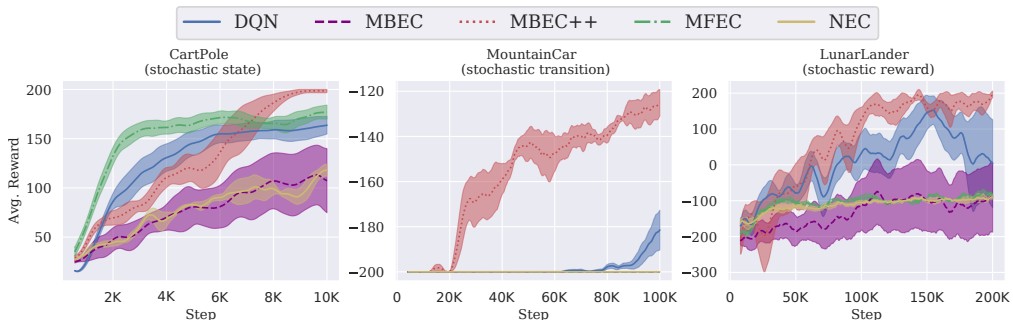

Figure 3: Average reward over learning steps on representative stochastic classical control environments (higher is better, mean and std. over 10 runs).

## 4.1 Grid-world: 2D Maze Exploration

We begin with simple maze navigation to explore scenarios wherein our proposed memory shows advantages. In this task, an agent is required to move from the starting point $(0, 0)$ to the endpoint $(n_e - 1, n_e - 1)$ in a maze environment of size $n_e \times n_e$. In the maze task, if the agent hits the wall of the maze, it gets $-1$ reward. If it reaches the goal, it gets $+1$ reward. For each step in the maze, the agent get $-0.1/n_e^2$ reward. An episode ends either when the agent reaches the goal or the number of steps exceeds 1000. We create different scenarios for this task ($n_e = 3$): noisy, trap and dynamic modes wherein our MBEC is compared with both parametric (DQN) and memory-based (MFEC; [4]) models (see Appendix B.2 for details and more results).

**Noisy mode** In this mode, the state is represented as the location plus an image of the maze. The image is preprocessed by a pretrained ResNet, resulting in a feature vector of $512$ dimensions (the output layer before softmax). We apply dropout to the image vector with different noise levels. We hypothesize that aggregating states into trajectory representation as in MBEC is a natural way to smooth out the noise of individual states.

Fig. 2 (a) measures sample efficiency of the models on noisy mode. Without noise, all models can quickly learn the optimal policy and finish 100 episodes within 1000 environment steps. However, as the increased noise distracts the agents, MFEC cannot find a way out until the episode ends. DQN performance reduces as the noise increases and ends up even worse than random exploration. By contrast, MBEC tolerates noise and performs much better than DQN and the random agent.

**Trap mode** The state is the position of the agent plus a trap location randomly determined at the beginning of the episode. If the agent falls into the trap, it receives a $-2$ reward (the episode does not terminate). This setting illustrates the advantage of memory-based planning. With MBEC, the agent remembers to avoid the trap by examining the future trajectory to see if it includes the trap. Estimating state-action value (DQN and MFEC) introduces overhead as they must remember all state-action pairs triggering the trap.

We plot the average reward and number of completed episodes over time in Fig. 2 (b). In this mode, DQN always learns a sub-optimal policy, which is staying in the maze. It avoids hitting the wall and trap, however, completes a relatively low number of episodes. MFEC initially learns well, quickly completing episodes. However its learning becomes unstable as more observations are written into its memory, leading to a lower performance over time. MBEC alone demonstrates stable learning, significantly outperforming other baselines in both reward and sample efficiency.

**Dynamic mode** The state is represented as an image and the maze structure randomly changes for each episode. A learnable CNN is equipped for each baseline to encode the image into a 64-dimensional vector. In this dynamic environment, similar state-actions in different episodes can lead to totally different outcomes, thereby highlighting the importance of trajectory modeling. In this case, MFEC uses VAE-CNN, trained to reconstruct the image state. Also, to verify the contribution of TR loss, we add two baselines: (i) MBEC without training trajectory model (no TRL) and (ii) MBEC with a traditional model-based transition prediction loss (TPL) (see Appendix B.2 for more details).

We compare the models' efficiency improvement over random policy by plotting the percentage of difference between the models' number of finished episodes and that of random policy in Fig. 2 (c). DQN and MFEC perform worse than random. MBEC with untrained trajectory model performs

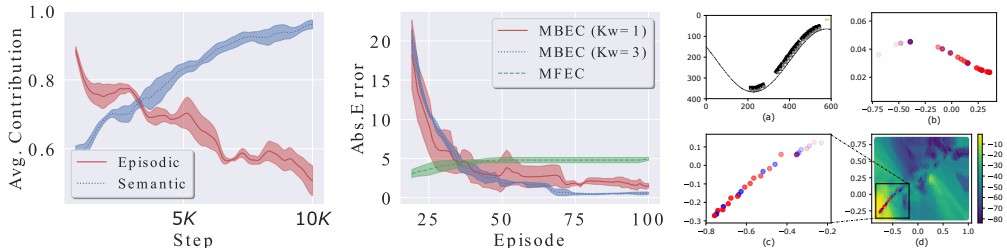

Figure 4: Cart Pole. (Left) Average contribution of episodic and semantic value estimations over timesteps (see Appendix B.3 for contribution definition). The episodic influence is gradually replaced by the semantic's. (Middle) The difference (absolute error) between the stored and true value of the starting state (mean and std. over 5 runs). (Right) Mountain Car. (a) Visualization of the car moving uphill over 30 timesteps. Due to noise, the next state can be observed as the current state with probability $p_{tr}$. (b) State and (c) trajectory spaces: the axes are the dimension of the trajectory vectors $\vec{\tau}$ projected into 2d space. Blue denotes the representation at noisy timestep and red the normal ones. Fading color denotes earlier timesteps. (d) Value estimation by the episodic memory for the whole 2-d trajectory space.

poorly. MBEC with trajectory model trained with TPL shows better performance than random, yet still underperforms our proposed MBEC with TRL by around 5-10%.

## 4.2 Stochastic Classical Control

In stochastic environments, taking correct actions can still lead to low rewards due to noisy observations, which negatively affects the quality of episodic memory. We consider 3 classical problems: Cart Pole, Mountain Car and Lunar Lander. For each problem, we examine RL agents in stochastic settings by (i) perturbing the reward with Gaussian (mean 0, std. $\sigma_{re} = 0.2$) or (ii) Bernoulli noise (with probability $p_{re} = 0.2$, the agent receives a reward $-r$ where $r$ is the true reward) and (iii) noisy transition (with probability $p_{tr} = 0.5$, the agent observes the current state as its next state despite taking any action). In this case, we compare MBEC++ with DQN, MFEC and NEC [36].

Fig. 3 shows the performances of the models on representative environments (full results in Appendix B.3). For easy problems like Cart Pole, although MFEC learns quickly, its over-optimistic control is sub-optimal in non-deterministic environments, and thus cannot solve the task. For harder problems, stochasticity makes state-based episodic memories (MFEC, NEC) fail to learn. DQN learns in these settings, albeit slowly, illustrating the disadvantage of not having a reliable episodic memory. Among all, MBEC++ is the only model that can completely solve the noisy Cart Pole within 10,000 steps and demonstrates superior performance in Mountain Car and Lunar Lander. Compared to MBEC++, MBEC performs badly, showing the importance of CLS.

**Behavior analysis** The episodic memory plays a role in MBEC++'s success. In the beginning, the agent mainly relies on the memory, yet later, it automatically learns to switch to the semantic value estimation (see Fig. 4 (left)). That is because in the long run the semantic value is more reliable and already fused with the episodic value through Eq. 7-8. Our `write` operator also helps MBEC++ in quickly searching for the optimal value. To illustrate that, we track the convergence of the episodic memory's written values for the starting state of the stochastic Cart Pole problem under a fixed policy (Fig. 4 (middle)). Unlike the over-optimistic MFEC's writing rule using `max` operator, ours enables mean convergence to the true value despite Gaussian reward noise. When using a moderate $K_w > 1$, the estimated value converges better as suggested by our analysis in Appendix A.2. Finally, to verify the contribution of the trajectory model, we examine MBEC++'s ability to counter noisy transition by visualizing the trajectory spaces for Mountain Car in Fig. 4 (right). Despite the noisy states (Fig. 4 (right, b)), the trajectory model can still roughly estimate the trace of trajectories (Fig. 4 (right, c)). That ensures when the agent approaches the goal, the trajectory vectors smoothly move to high-value regions in the trajectory space (Fig. 4 (right, d)). We note that for this problem that ability is only achieved through training with TR loss (comparison in Appendix B.3).

**Ablation study** We pick the noisy transition Mountain Car problem for ablating components and hyper-parameters of MBEC++ with different neighbors ($K$), chunk length ($L$) and memory slots ($N$). The results in Fig. 5 demonstrate that the performance improves as $K$ increases, which is

| Model | All | 25 games |
|---|---|---|
| Nature DQN | 15.7/51.3 | 83.6/16.0 |
| MFEC | 85.0/45.4 | 77.7/40.9 |
| NEC | 99.8/54.6 | 106.1/53.3 |
| EMDQN* | 528.4/92.8 | 250.6/95.5 |
| EVA | - | 172.2/39.2 |
| ERLAM | - | 515.4/103.5 |
| MBEC++ | **654.0/117.2** | **518.2/133.4** |

Table 1: Human normalized scores (mean/median) at 10 million frames for all and a subset of 25 games. Baselines' numbers are adopted from original papers and [46], respectively. * The baseline is reported with 40 million frames of training. - The exact numbers are not reported.

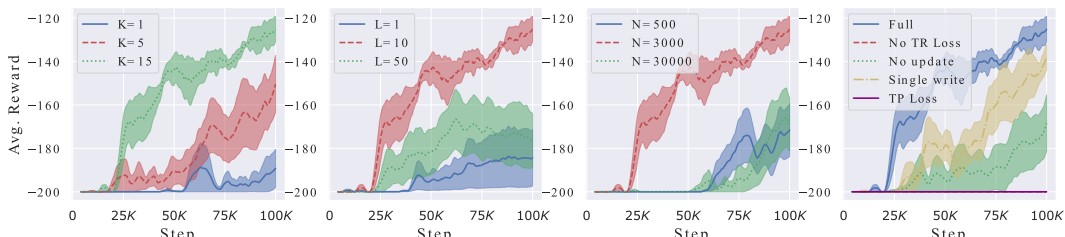

Figure 5: Noisy Transition Mountain Car: ablation study. Each plot varies one hyper-parameter or ablated component while fixing others in default values ($K = 15$, $L = 10$, $N = 3000$).

common for KNN-based methods. We also find that using a too short or too long chunk deteriorates the performance of MBEC++. Short chunk length creates redundancy as the stored trajectories will be similar while long chunk length makes minimizing TR loss harder. Finally, the results confirm that the learning of MBEC++ is hindered significantly with small memory. A too-big memory does not help either since the trajectory model continually refines the trajectory representation, a too big memory slows the replacement of old representations with more accurate newer ones.

We also ablate MBEC++: (i) without TR loss, (ii) with TP loss (iii), without multiple write ($K_w = 1$) and (iv) without memory update. We realize that the first two configurations show no sign of learning. The last two can learn but much slower than the full MBEC++, justifying our neighbor memory writing and update (Fig. 5 (rightmost)). More ablation studies are in Appendix B.6 where we find our dynamic consolidation is better than fixed combinations and optimal $p_{read}$ is 0.7.

## 4.3 Atari 2600 Benchmark

We benchmark MBEC++ against other strong episodic memory models in playing Atari 2600 video games [3]. The task can be challenging with stochastic and partially observable games [23]. Our model adopts DQN [34] with the same setting (details in Appendix B.4). We only train the models within 10 million frames for sample efficiency.

Table 1 reports the average performance of MBEC++ and baselines on all (57) and 25 popular Atari games concerning human normalized metrics [34]. Compared to the vanilla DQN, MFEC and NEC, MBEC++ is significantly faster at achieving high scores in the early learning phase. Further, MBEC++ outperforms EMDQN even trained with 40 million frames and achieves the highest median score. Here, state-action value estimations fall short in quickly solving complicated tasks with many actions like playing Atari games as it takes time to visit enough state-action pairs to create useful memory's stored values. By contrast, when the models in MBEC++ are well-learnt (which is often within 5 million frames, see Appendix B.4), its memory starts providing reliable trajectory value estimation to guide the agent to good policies. Remarkably, our episodic memory is much smaller than that of others and our trajectory model size is insignificant to DQN's (Appendix B.4 and Table 2).

In the subset testbed, MBEC++ demonstrates competitive performance against trajectory-utilized models including EVA [16] and ERLAM [46]. These baselines work with trajectories as raw state-action sequences, unlike our distributed trajectories. In the mean score metric, MBEC++ is much better than EVA (nearly double score) and slightly better than ERLAM. MBEC++ agent

plays consistently well across games without severe fluctuations in performance, indicated by its significantly higher median score.

We also compare MBEC++ with recent model-based RL approaches including Dreamer-v2 [15] and SIMPLE [23]. The results show that our method is competitive against these baselines. Notably, our trajectory model is much simpler than the other methods (we only have TR and reward losses and our network is the standard CNN of DQN for Atari games). Appendix B.4 provides more details, learning curves and further analyses.

### 4.4 POMDP: 3D Navigation

To examine MBEC++ on Partially-Observable Markov Decision Process (POMDP) environments, we conduct experiments on a 3D navigation task: Gym Mini-World's Pickup Objects [5]. Here, an agent moves around a big room to collect several objects (+1 reward for each picked object). The location, shape and color of the objects change randomly across episodes. The state is the frontal view-port of the agent and encoded by a common CNN for all baselines (details in Appendix B.5).

We train all models for only 2 million steps and report the results for a different number of objects in Appendix's Fig. 12. Among all baselines, MBEC++ demonstrates the best learning progress and consistently improves over time. Other methods either fail to learn (DRQN) or show a much slower learning speed (DQN and PPO). That proves our MBEC++ is useful for POMDP.

## 5   Conclusion

We have introduced a new episodic memory that significantly accelerates reinforcement learning in various problems beyond near-deterministic environments. Its success can be attributed to: (a) storing distributed trajectories produced by a trajectory model, (b) memory-based planning with fast value-propagating memory writing and refining, and (c) dynamic consolidation of episodic values to parametric value function. Our experiments demonstrate the superiority of our method to prior episodic controls and strong RL baselines. One limitation of this work is the large number of hyperparameters, which prevents us from fully tuning MBEC++. In future work, we will extend to continuous action space and explore multi-step memory-based planning capability of our approach.

Our research aims to improve sample-efficiency of RL and can be trained with common computers. Our method improves the performance in various RL tasks, and thus opens the chance for creating better autonomous systems that work flexibly across sectors (robotics, manufacturing, logistics, and decision support systems). Although we do not think there are immediate bad consequences, we are aware of potential problems. First, our method does not guarantee safe exploration during training. If learning happens in a real-world setting (e.g. self-driving car), the agent can make unsafe exploration (e.g. causing accidents). Second, we acknowledge that our method, like many other Machine Learning algorithms, can be misused in unethical or malicious activities.

**ACKNOWLEDGMENTS**

This research was partially funded by the Australian Government through the Australian Research Council (ARC). Prof Venkatesh is the recipient of an ARC Australian Laureate Fellowship (FL170100006).

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
