# Appendix

## A  Analytical Studies on Model-based Episodic Memory

### A.1  Why Is Trajectorial Recall (TR) Loss Good for Episodic Memory?

For proper episodic control, neighboring keys should represent similar trajectories. If we simply assume that two trajectories are similar if they share many common transitions, training the trajectory model with TR loss indeed somehow enforces that property. To illustrate, we consider simple linear $\mathcal{T}_\phi$ and $\mathcal{G}_\omega$ such that the reconstruction process becomes

$$y^* (t) = W \left( U \overrightarrow{\tau}_t + V \left[ s_{t'}, a_{t'} \right] \right)$$

Here, we also assume that the query is clean without added noise. Then we can rewrite TR loss for a trajectory $\tau_t$

$$\mathcal{L}_{tr} (\tau_t) = \sum_{t'=1}^{t} \left\| W \left( U \overrightarrow{\tau}_t + V \left[ s_{t'}, a_{t'} \right] \right) - \left[ s_{t'+1}, a_{t'+1} \right] \right\|_2^2$$

$$= \sum_{t'=1}^{t} \left\| \Delta_{t'} (\tau_t) \right\|_2^2$$

Let us denote $S \neq \emptyset$ the set of common transition steps between 2 trajectories: $\tau_{t_1}^1$ and $\tau_{t_2}^2$, by applying triangle inequality,

$$\mathcal{L}_{tr} \left( \tau_{t_1}^1 \right) + \mathcal{L}_{tr} \left( \tau_{t_2}^2 \right) \geq \sum_{t' \in S} \left\| \Delta_{t'} \left( \tau_{t_1}^1 \right) \right\|_2^2 + \left\| \Delta_{t'} \left( \tau_{t_2}^2 \right) \right\|_2^2$$

$$\geq \sum_{t' \in S} \left\| \Delta_{t'} \left( \tau_{t_1}^1 \right) - \Delta \left( \tau_{t_2}^2 \right) \right\|_2^2$$

$$= |S| \left\| WU \left( \overrightarrow{\tau}_{t_1}^1 - \overrightarrow{\tau}_{t_2}^2 \right) \right\|_2^2$$

If we assume $WU \left( \overrightarrow{\tau}_{t_1}^1 - \overrightarrow{\tau}_{t_2}^2 \right) \neq 0$ as $\overrightarrow{\tau}_{t_1}^1 \neq \overrightarrow{\tau}_{t_2}^2$, applying Lemma 2.3 in [12] yields

$$\frac{\mathcal{L}_{tr} \left( \tau_{t_1}^1 \right) + \mathcal{L}_{tr} \left( \tau_{t_2}^2 \right)}{|S| \, \sigma_{min} (WU)} \geq \left\| \overrightarrow{\tau}_{t_1}^1 - \overrightarrow{\tau}_{t_2}^2 \right\|_2^2$$

where $\sigma_{min} (WU)$ is the smallest nonzero singular value of $WU$. As the TR loss decreases and the number of common transition increases, the upper bound of the distance between two trajectory vectors decreases, which is desirable. On the other hand, it is unclear whether the traditional transition prediction loss holds that property.

### A.2  Convergence Analysis of write Operator

In this section, we show that we can always find $\alpha_w$ such that the writing converges with probability 1 and analyze the convergence as $\alpha_w$ is constant. To simplify the notation, we rewrite Eq. 2 as

$$\mathcal{M}_i^v (n+1) = \mathcal{M}_i^v (n) + \lambda (n) \left( R_j (n) - \mathcal{M}_i^v (n) \right) \tag{9}$$

where $i$ and $j$ denote the current memory slot being updated and its neighbor that initiates the writing, respectively. $\lambda (n) = \alpha_w (n) \frac{\langle \rangle_{ij} (n)}{\sum_{b \in \mathcal{N}_j^{K_w}} \langle \rangle_{bj} (n)}$ where $\mathcal{N}_j^{K_w}$ is the set of $K_w$ neighbors of $j$. $R_j$ is the empirical return of the trajectory whose key is the memory slot $j$, $\langle \rangle_{ij}$ the kernel function of 2 keys and $n$ the number of updates. As mentioned in [40], this stochastic approximation converges when $\sum_{n=1}^{\infty} \lambda (n) = \infty$ and $\sum_{n=1}^{\infty} \lambda^2 (n) < \infty$.

By definition, $\langle\rangle_{ij} = \frac{1}{\|\vec{\tau}_i - \vec{\tau}_j\| + \epsilon}$ and $\|\vec{\tau}\| \leq 1$ since $\vec{\tau}$ is the hidden state of an LSTM. Hence, we have $\forall i, j : 0 < \frac{1}{2+\epsilon} \leq \langle\rangle_{ij} \leq \frac{1}{\epsilon}$. Hence, let $B_{ij}(n)$ a random variable denoting $\frac{\langle\rangle_{ij}(n)}{\sum_{b \in \mathcal{N}_j^{K_w}} \langle\rangle_{bj}(n)}$–the neighbor weight at step $n$, $\forall i, j$ :

$$\frac{\epsilon}{K_w \epsilon + 2K_w - 2} \leq B_{ij}(n) \leq \frac{2 + \epsilon}{K_w \epsilon + 2}$$

That yields $\sum_{n=1}^{\infty} \lambda(n) \geq \frac{\epsilon}{K_w \epsilon + 2K - 2} \sum_{n=1}^{\infty} \alpha_w(n)$ and $\sum_{n=1}^{\infty} \lambda^2(n) \leq \left(\frac{2+\epsilon}{K_w \epsilon + 2}\right)^2 \sum_{n=1}^{\infty} \alpha_w^2(n)$. Hence the writing updates converge when $\sum_{n=1}^{\infty} \alpha_w(n) = \infty$ and $\sum_{n=1}^{\infty} \alpha_w^2(n) < \infty$. We can always choose such $\alpha_w$ (e.g., $\alpha_w(n) = \frac{1}{n+1}$).

With a constant writing rate $\alpha$, we rewrite Eq. 9 as

$$
\begin{aligned}
\mathcal{M}_i^v(n+1) &= \mathcal{M}_i^v(n) + \alpha B_{ij}(n)(R_j(n) - \mathcal{M}_i^v(n)) \\
&= \alpha B_{ij}(n) R_j(n) + \mathcal{M}_i^v(n)(1 - \alpha B_{ij}(n)) \\
&= \sum_{t=1}^{n} \alpha B_{ij}(t) \prod_{l=t+1}^{n} (1 - \alpha B_{ij}(l)) R_j(t) \\
&\quad + \prod_{t=1}^{n} (1 - \alpha B_{ij}(t)) \mathcal{M}_i^v(1)
\end{aligned}
$$

where the second term $\prod_{t=1}^{n} (1 - \alpha B_{ij}(t)) \mathcal{M}_i^v(1) \to 0$ as $n \to \infty$ since $B_{ij}(t)$ and $\alpha$ are bounded between 0 and 1. The first term can be decomposed into three terms

$$\sum_{t=1}^{n} \alpha B_{ij}(t) \prod_{l=t+1}^{n} (1 - \alpha B_{ij}(l)) R_j(t) = T_1 + T_2 + T_3$$

where

$$
\begin{aligned}
T_1 &= \sum_{t=1}^{n} \alpha B_{ij}(t) \prod_{l=t+1}^{n} (1 - \alpha B_{ij}(l)) V_i \\
T_2 &= \sum_{t=1}^{n} \alpha B_{ij}(t) \prod_{l=t+1}^{n} (1 - \alpha B_{ij}(l)) \Delta V_{ij}(t) \\
T_3 &= \sum_{t=1}^{n} \alpha B_{ij}(t) \prod_{l=t+1}^{n} (1 - \alpha B_{ij}(l)) \tilde{R}_j(t)
\end{aligned}
$$

Here, $V_i$ is the true value of the trajectory stored in slot $i$, $\Delta V_{ij}(t) = V_j(t) - V_i$ and $\tilde{R}_j(t) = R_j(t) - V_j(t)$ the noise term between the return and the true value. Assume that the value is associated with zero mean noise and the value noise is independent with the neighbor weights, then $\mathbb{E}(T_3) = 0^2$.

Further, we make other two assumptions: (1) the neighbor weights are independent across update steps; (2) the probability $p_j$ of visiting a neighbor $j$ follows the same distribution across update steps and thus, $\mathbb{E}(B_{ij}(t)) = \mathbb{E}(B_{ij}(l)) = \mathbb{E}(B_{ij})$. We now can compute

---

[2]This assumption is true for the Perturbed Cart Pole Gaussian reward noise.

$$\mathbb{E}\left(T_1\right) = \mathbb{E}\left(\sum_{t=1}^{n} \alpha B_{ij}\left(t\right) \prod_{l=t+1}^{n} \left(1 - \alpha B_{ij}\left(l\right)\right) V_i\right)$$

$$= V_i \sum_{t=1}^{n} \alpha \mathbb{E}\left(B_{ij}\right) \prod_{l=t+1}^{n} \left(1 - \alpha \mathbb{E}\left(B_{ij}\right)\right)$$

$$= V_i \alpha \mathbb{E}\left(B_{ij}\right) \sum_{t=1}^{n} \left(1 - \alpha \mathbb{E}\left(B_{ij}\right)\right)^{n-t}$$

$$= V_i \alpha \mathbb{E}\left(B_{ij}\right) \frac{1 - \left(1 - \alpha \mathbb{E}\left(B_{ij}\right)\right)^{n}}{1 - \left(1 - \alpha \mathbb{E}\left(B_{ij}\right)\right)}$$

$$= V_i \left(1 - \left(1 - \alpha \mathbb{E}\left(B_{ij}\right)\right)^{n}\right)$$

As $n \to \infty$, $\mathbb{E}\left(T_1\right) \to V_i$ since since $B_{ij}\left(t\right)$ and $\alpha$ are bounded between 0 and 1.

Similarly, $\mathbb{E}\left(T_3\right) = \mathbb{E}\left(V_j\left(t\right) - V_i\right) = \mathbb{E}\left(V_j\left(t\right)\right) - V_i = \sum_{j \in \mathcal{N}_i^{K_w}} p_j V_j - V_i$, which is the approximation error of the KNN algorithm. Hence, with constant learning rate, on average, the write operator leads to the true value plus the approximation error of KNN. The quality of KNN approximation determines the mean convergence of write operator. Since the bias-variance trade-off of KNN is specified by the number of neighbors $K$, choosing the right $K > 1$ (not too big, not too small) is important to achieve good writing to ensure fast convergence. That explains why our writing to multiple slots ($K > 1$) is generally better than the traditional writing to single slot ($K = 1$).

### A.3   Convergence Analysis of refine Operator

In this section, we study the convergence of the memory-based value estimation by applying refine operator to the memory. As such, we treat the $\mathrm{read}\left(\overrightarrow{\tau}|\mathcal{M}\right)$ operator as a value function over trajectory space $\mathcal{T}$ and simplify the notation as $\mathrm{read}\left(x\right)$ where $x$ represents the trajectory. We make the assumption that the read operator simply uses averaging rule and the set of neighbors stored in the memory is fixed (i.e. no new element is added to the memory) , then

$$\mathrm{read}_t\left(x\right) \quad = \quad \sum_{i \in \mathcal{N}^K\left(x\right)} B_{ix} \mathcal{M}_{i,t}^v$$

where $B_{ix} = \frac{\left\langle \mathcal{M}_i^k, x\right\rangle}{\sum_{j \in \mathcal{N}^K\left(x\right)} \left\langle \mathcal{M}_j^k, x\right\rangle}$ is the neighbor weight and $t$ is the step of updating.

We rewrite the refine operator as

$$\mathcal{M} \leftarrow \mathrm{write}\left(x, \max_a r_\varphi\left(x, a\right) + \gamma \mathrm{read}_t\left(y\right)|\mathcal{M}\right)$$
$$\Leftrightarrow \forall i \in \mathcal{N}^K\left(x\right):$$
$$\mathcal{M}_{i,t+1}^v = \mathcal{M}_{i,t}^v$$
$$+ \alpha_{w,t} B_{ix}\left(\max_a r_\varphi\left(x, a\right) + \gamma \mathrm{read}_t\left(y\right) - \mathcal{M}_{i,t}^v\right)$$

where $y$ is the trajectory after taking action $a$ from the trajectory $x$. Then, after the refine,

$$\text{read}_{t+1}(x) = \sum_{i \in \mathcal{N}^K(x)} B_{ix} \mathcal{M}^v_{i,t+1}$$

$$= \sum_{i \in \mathcal{N}^K(x)} \mathcal{M}^v_{i,t} B_{ix} (1 - \alpha_{w,t})$$

$$+ \alpha_{w,t} \sum_{i \in \mathcal{N}^K(x)} \left( \max_a r_\varphi(x,a) + \gamma \text{read}_t(y) \right) B^2_{ix}$$

$$+ \alpha_{w,t} \sum_{i \in \mathcal{N}^K(x)} \mathcal{M}^v_{i,t} B_{ix} (1 - B_{ix})$$

$$= \text{read}_t(x)(1 - \alpha_{w,t}) + \alpha_{w,t} G_t(x)$$

where $G_t(x) = \max_a U_t(x,a) \sum_{i \in \mathcal{N}^K(x)} B^2_{ix} + \sum_{i \in \mathcal{N}^K(x)} \mathcal{M}^v_{i,t} B_{ix} (1 - B_{ix})$, $U(x,a) = r_\varphi(x,a) + \gamma \text{read}(y)$. To simplify the analysis, we assume the stored neighbors of $x$ are apart from $x$ by the same distance, i.e., $\forall i \in \mathcal{N}^K(x) : B_{ix} = \frac{1}{K}$. That is,

$$G(x) = \left( \max_a r_\varphi(x,a) + \gamma \text{read}(y) \right) \frac{1}{K} + \text{read}(x) \frac{K-1}{K}$$

Let H an operator defined for the function read : $\mathcal{T} \to \mathbb{R}$ as

$$\text{Hread}(x) = \sum_{\hat{y} \in \mathcal{T}} P_{a^*}(x,y) G_t(x|a^*)$$

where $a^* = \operatorname*{argmax}_a \sum_{\hat{y} \in \mathcal{T}} P_a(x,y) U(x,a)$. We will prove H is a contraction in the sup-norm.

Let us denote $\Delta \text{read}(x) = \text{read}_1(x) - \text{read}_2(x)$, $\Delta PU(x, a^*_1, a^*_2) = \sum_{y \in \mathcal{T}} P_{a^*_1}(x,y) U_1(x, a^*_1) - \sum_{y \in \mathcal{T}} P_{a^*_2}(x,y) U_2(x, a^*_2)$ and $\hat{a} = \operatorname*{argmax}_{a^*_1, a^*_2} \sum_{y \in \mathcal{T}} P_{a^*_1}(x,y) U_1(x, a^*_1)$, $\sum_{y \in \mathcal{T}} P_{a^*_2}(x,y) U_2(x, a^*_2)$. Then,

| Task | MBEC | MBEC++ | DQN |
|------|------|--------|-----|
| 2D Maze | 2K | N/A | 43K |
| Classical control | N/A | 39K | 43K |
| Atari games | N/A | 13M | 13M |
| 3D Navigation | N/A | 13M | 13M |

Table 2: The number of trainable parameters of MBEC(++) and its main competitor DQN.

$$\|\text{Hread}_1 - \text{Hread}_2\|_\infty = \left\| \sum_{y \in \mathcal{T}} \left( \Delta PU \left(x, a_1^*, a_2^*\right) \frac{1}{K} \right. \right.$$

$$\left. \left. + \ \Delta\text{read}\left(x\right) \frac{K-1}{K} \right) \right\|_\infty$$

$$\leq \left\| \sum_{y \in \mathcal{T}} \Delta PU \left(x, a_1^*, a_2^*\right) \frac{1}{K} \right\|_\infty$$

$$+ \left\| \frac{K-1}{K} \Delta\text{read}\left(x\right) \right\|_\infty$$

$$\leq \left\| \sum_{y \in \mathcal{T}} \frac{P_{\hat{a}}\left(x, y\right)}{K} \gamma \Delta\text{read}\left(y\right) \right\|_\infty$$

$$+ \left\| \frac{K-1}{K} \Delta\text{read}\left(x\right) \right\|_\infty$$

$$\leq \sum_{y \in \mathcal{T}} \frac{P_{\hat{a}}\left(x, y\right)}{K} \gamma \left\| \Delta\text{read}\left(y\right) \right\|_\infty$$

$$+ \left\| \frac{K-1}{K} \Delta\text{read} \right\|_\infty$$

$$\leq \frac{\gamma + K - 1}{K} \left\| \text{read}_1 - \text{read}_2 \right\|_\infty$$

Since $\gamma < 1$, $0 < \gamma_K = \frac{\gamma+K-1}{K} < 1 \ \forall K \geq 1$. Thus, H is a contraction in the sup-norm and there exists a fix-point $\text{read}^*$ such that $\text{Hread}^* = \text{read}^*$.

We define $\Delta_t = \text{read}_t - \text{read}^*$, then

$$\Delta_{t+1} = \Delta_t\left(x\right)\left(1 - \alpha_{w,n}\right) + \alpha_{w,t} F_t\left(x\right)$$

where $F_t\left(x\right) = G_t\left(x\right) - \text{read}^*\left(x\right)$. We have

$$\mathbb{E}\left(F_t\left(x\right) \mid F_t\right) = \sum_{\hat{y} \in \mathcal{T}} P_{a^*}\left(x, y\right) G_t\left(x|a^*\right) - \text{read}^*\left(x\right)$$

$$= \text{Hread}_t\left(x\right) - \text{read}^*\left(x\right)$$

Following the proof in [32], $\mathbb{E}\left(F_t\left(x\right) \mid F_t\right) \leq \gamma_K \left\|\Delta_t\left(x\right)\right\|_\infty$ and $\text{var}\left(F_t\left(x\right) \mid F_t\right) < C\left(1 + \left\|\Delta_t\left(x\right)\right\|\right)^2$ for $C > 0$. Assume that $\sum_{t=1}^\infty \alpha_{w,t} = \infty$ and $\sum_{t=1}^\infty \alpha_{w,t}^2 < \infty$, according to [22], $\Delta_t$ converges to 0 with probability 1 or $\text{read}$ converges to $\text{read}^*$.

## B  Experimental Details

### B.1  Implemented baseline description

In this section, we describe baselines that are implemented and used in our experiments. We trained all the models using a single GPU Tesla V100-SXM2.

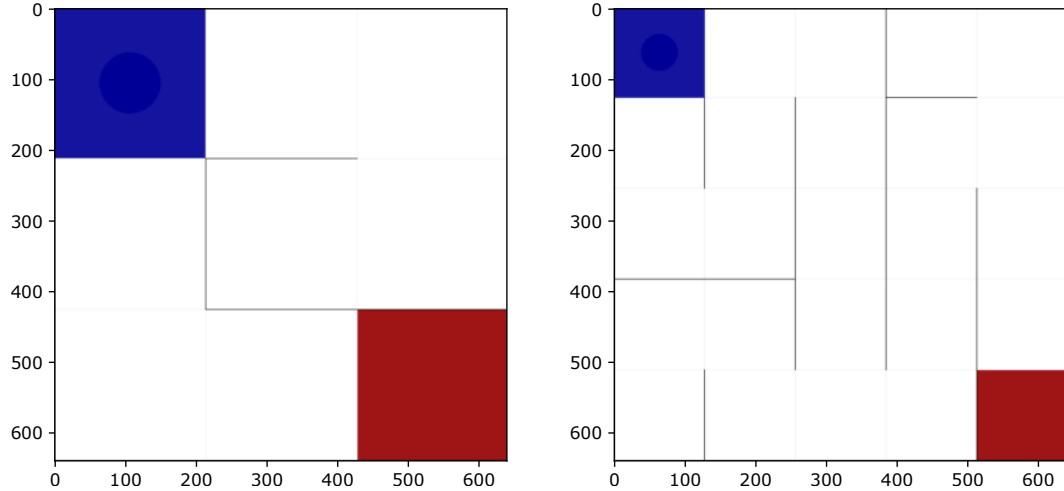

Figure 6: Maze map $3 \times 3$ (left) and $5 \times 5$ (right). The agent starts from the left top corner (blue square) and finds the goal at the right bottom corner (red square).

**Model-based Episodic Control (MBEC, ours)**    The main hyper-parameter of MBEC is the number of neighbors ($K$), chunk length ($L$) and memory slots ($N$). Hyper-parameter tuning is adjusted according to specific tasks. For example, for small and simple problems, $K$ and $N$ tend to be smaller and $L$ is often about $20-30\%$ of the average length per episode. Across experiments, we follow prior works using $\gamma = 0.99$. We also fix $\alpha_w = 0.5$ to reduce hyperparameter tuning. To implement read and write, we set $K = K_r = K_w$ and use the kernel $\langle x, y \rangle = \frac{1}{\|x-y\|+\epsilon}$ with $\epsilon = 10^{-3}$ following [36].

Unless stated otherwise, the hidden size of the trajectory model is fixed to $H = 16$ for all tasks. The reward model is implemented as a 2-layer ReLU feed-forward neural network and trained with batch size 32 for all tasks. To compute TR Loss, we sample 4 past transitions and add Gaussian noise (mean 0, std. $0.1 \times \|q\|_2$) to the query vector $q$. Notably, in MBEC++, when training with TD loss, we do not back-propagate the gradient to the trajectory model to ensure that the trajectory representations are only shaped by the TR Loss.

In practice, to reduce computational complexity, we do not perform refine operator every timestep. Rather, at each step, we randomly refine with a probability $p_u = 0.1$. Similarly, we occasionally update the parameters of the trajectory model. Every $L$ step, we randomly update $\phi$ and $\omega$ using back-propagation via $\mathcal{L}_{tr}$ with probability $p_{rec} = 0.5$. For Atari games, we stop training the trajectory model after 5 million steps. On our machine for Atari games, these tricks generally make MBEC++ run at speed 100 steps/s while DQN 150 steps/s.

**Deep Q-Network (DQN)**    Except for Sec. 4.3 and 4.4, we implement DQN[3] with the following hyper-parameters: 3-layer ReLU feed-forward Q-network (target network) with hidden size 144, target update every 100 steps, TD update every 1 step, replay buffer size $10^6$ and Adam optimizer with batch size 32. The exploration rate decreases from 1 to 0.01. We tune the learning rate for each task in range $\left[10^{-3}, 10^{-5}\right]$. For tasks with image input, the Q-network (target network) is augmented with CNN to process the image depending on tasks. MBEC++ adopts the same DQN with a smaller hidden size of 128. Table 2 compares model size between DQN and MBEC(++). Regarding memory usage, for Atari games, DQN consumes 1,441 MB and MBEC++ 1,620 MB.

**Model-Free Episodic Control (MFEC)**    This episodic memory maintains a value table using $K$-nearest neighbor to read the value for a query state and max operator to write a new value. We set the key dimension and memory size to 64 and $10^6$, respectively. We tune $K \in \{3, 5, 11, 25\}$ for each task. Unless stated otherwise, we use random projection for MFEC. For VAE-CNN version used in dynamic maze mode, we use 5-convolutional-layer encoder and decoder (16-128 kernels with 4×4 kernel size and a stride of 2). Other details follow the original paper [4].

---

[3]https://github.com/higgsfield/RL-Adventure

**Neural Episodic Control (NEC)**  This model extends MFEC using the state-key mapping as a CNN embedding network trained to minimize the TD error of memory-based value estimation. Also, multi-step Q-learning update is employed for memory writing. We adopt the publicly available source code [4] which follows the same hyper-parameters used in the original paper [36] and apply it to stochastic control problem by implementing the embedding network as a 2-layer feed-forward neural network. We tune $K \in \{3, 5, 11, 25\}$ and the hidden size of the embedding network $\in \{32, 64, 128, 256\}$ for each task.

**Proximal Policy Optimization (PPO)**  PPO [38] is a policy gradient method that simplifies Trust Region update with gradient descent and soft constraint (maintaining low KL divergence between new and old policy via objective clipping). We test PPO for the 3D Navigation task using the original source code of the environment Gym Mini World.

**Deep Recurrent Q-Network (DRQN)**  DRQN [17] is similar to DQN except that it uses LSTM as the Q-Network. As the hidden state of LSTM represents the environment state for the Q-Network, it captures past information that may be necessary for the agent in POMDP. We extend DQN to DRQN by storing transitions with the hidden states in the replay buffer and replacing the feed-forward Q-Network with an LSTM Q-Network. We tune the hidden size of the LSTM $\in \{128, 256, 512\}$ for 3D navigation task.

## B.2  Maze task

**Task overview**  In the maze task, if the agent hits the wall of the maze, it gets $-1$ reward. If it reaches the goal, it gets $1$ reward. For each step in the maze, the agent get $-0.1/n_e^2$ reward. An episode ends either when the agent reaches the goal or the number of steps exceeds 1000.

To build different modes of the task, we modify the original gym-maze environment[5]. Fig. 6 illustrates the original $3 \times 3$ and $5 \times 5$ maze structure. We train and tune MBEC and other baselines for $3 \times 3$ maze task and use the found hyper-parameters for other task modes. For MBEC, the best hyper-parameters are $K = 5$, $L = 5$, $N = 1000$.

**Transition Prediction (TP) loss**  For dynamic mode and ablation study for stochastic control tasks, we adopt a common loss function to train the traditional model in model-based RL: the transition prediction (TP) loss. Trained with the TP loss, the model tries to predict the next observations given current trajectory and observations. The TP loss is concretely defined as follows,

$$\mathcal{L}_{tp} = \mathbb{E}\left(\|y^*(t) - [s_{t+1}, a_{t+1}]\|_2^2\right) \tag{10}$$

$$y^*(t) = \mathcal{G}_\omega\left(\mathcal{T}_\phi\left([\tilde{s}_t, \tilde{a}_t], \overrightarrow{\tau}_{t-1}\right)\right) \tag{11}$$

The key difference between TP loss and TR loss is the timestep index. TP loss takes observations at current timestep to predict the one at the next timestep. On the other hand, TR loss takes observations at past timestep and uses the current working memory (hidden state of the LSTM) to reconstruct the observations at the timestep after the past timestep. Our experiments consistently show that TP loss is inferior to our proposed TR loss (see Sec. B.3).

## B.3  Stochastic classical control task

**Task description**  We introduce three ways to make a classical control problem stochastic. First, we add Gaussian noise (mean 0, $\sigma_{re} = 0.2$) to the reward that the agent observes. Second, we add Bernoulli-like noise (with a probability $p_{re} = 0.2$, the agent receives a reward $-r$ where $r$ is the true reward). Finally, we make the observed transition noisy by letting the agent observe the same state

---

[4] `https://github.com/hiwonjoon/NEC`
[5] `https://github.com/MattChanTK/gym-maze`

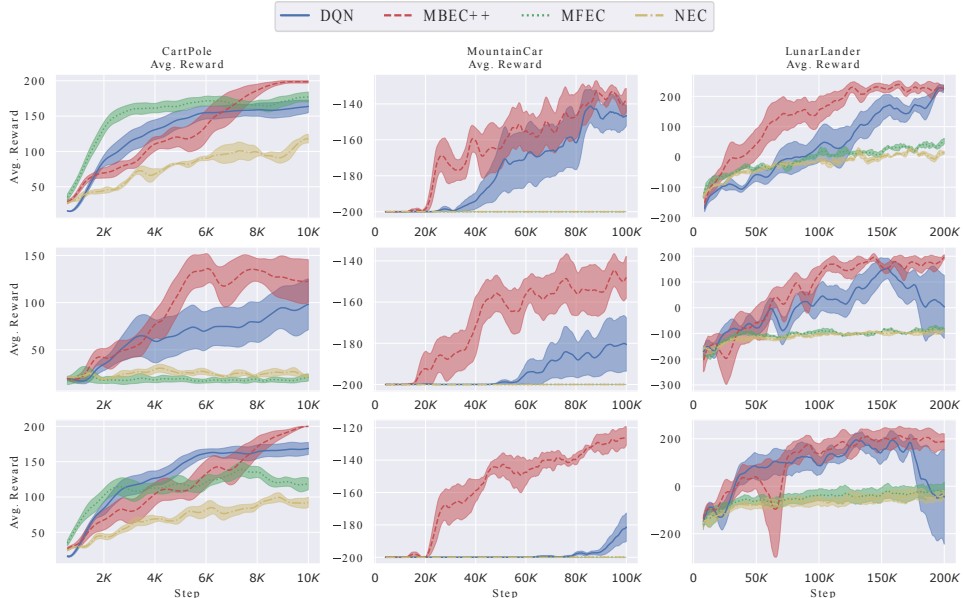

Figure 7: All learning curves for stochastic classical control task. First row: Gaussian noisy reward. Second row: Bernoulli noisy reward. Third row: Noisy transition

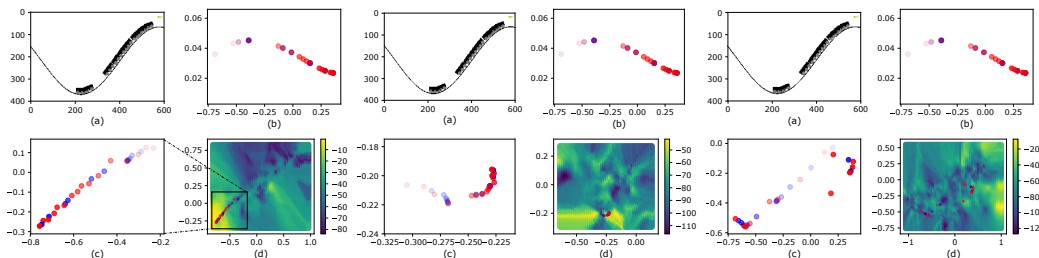

Figure 8: Stochastic Mountain Car. Trajectory space in noisy transition using models trained with TR loss (left), no training (middle) and TP loss (right).

despite taking any action with a probability $p_{tr} = 0.5$. The randomness only affects what the agent sees while the environment dynamic is not affected. Three classical control problems are chosen from Open AI's gym: CartPole-v0, MountainCar-v0 and LunarLander-v2. For each problem, we apply the three stochastic configurations, yielding 9 tasks in total.

Fig. 7 showcases the learning curves of DQN, MBEC++, MFEC and NEC for all 9 tasks. MBEC++ is consistently the leading performer. DQN is often the runner-up, yet usually underperforms our method by a significant margin. Overall, other memory-based methods such as MFEC and NEC perform poorly for these tasks since they are not designed for stochastic environments.

**Memory contribution** We determine the episodic and semantic contribution to the final value estimation by counting the number of times their greedy actions equal the final greedy action, dividing by the number of timesteps. Concretely, the episodic and semantic contribution is computed respectively as

$$\frac{\sum_{t=1}^{T} \operatorname{argmax}_{a_t} Q_{eps}(s_t, a_t) == \operatorname{argmax}_{a_t} Q(s_t, a_t)}{T}$$

$$\frac{\sum_{t=1}^{T} \operatorname{argmax}_{a_t} Q_{\theta}(s_t, a_t) == \operatorname{argmax}_{a_t} Q(s_t, a_t)}{T}$$

where $Q_{eps}(s_t, a_t) = Q_{MBEC}(s_t, a_t) f_{\beta}(s_t, \overrightarrow{\tau}_{t-1})$, $Q_{\theta}(s_t, a_t)$ and $Q(s_t, a_t)$ represent the episodic, semantic and final value estimation, respectively.

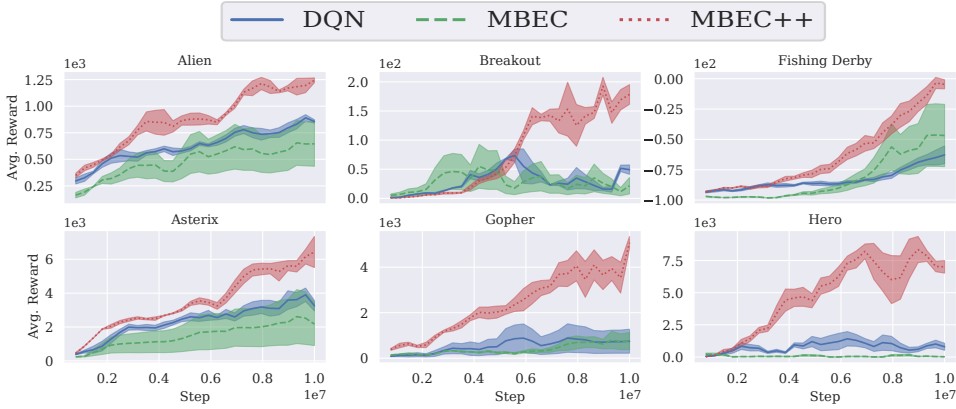

Figure 9: Learning curves of several Atari games.

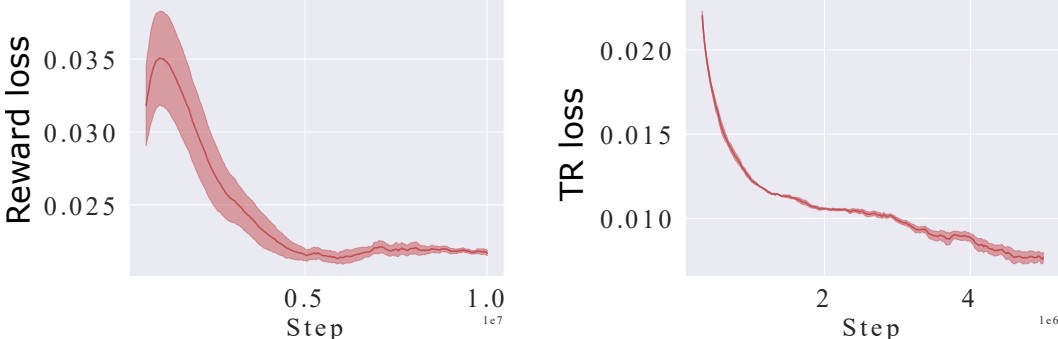

Figure 10: Loss of reward model and TR loss over training iterations for Atari's Asterix and Gopher, respectively.

Fig. 4 illustrates the running average contribution using a window of 100 timesteps. We note that the contribution of the two does not need to sum up to 1 as both can agree with the same greedy action.

**Trajectory space visualization** We visualize the memory-based value function w.r.t trajectory vectors in Fig. 4 (d). As such, we set the trajectory dimension to $H = 2$ and estimate the value for each grid point (step 0.05) using read operator as

$$V\left(\overrightarrow{\tau}\right) \approx \sum_{i \in \mathcal{N}^{K_r}(\overrightarrow{\tau})} \frac{\left\langle \mathcal{M}_i^k, \overrightarrow{\tau} \right\rangle \mathcal{M}_i^v}{\sum_{j \in \mathcal{N}^{K}(\overrightarrow{\tau})} \left\langle \mathcal{M}_j^k, \overrightarrow{\tau} \right\rangle}$$

To cope with noisy environment, MBEC relies on noise-tolerant trajectory representation. As demonstrated in Fig. 8, even when the state representations are disturbed by not changing to true states, the trajectory representations shaped by the TR loss maintain good approximation and interpolate well the latent location of disturbed trajectories. In contrast, representations generated by random model or model trained with TP loss fail to discover latent location of disturbed trajectories, either collapsing (random model) or shattering (TP loss model).

The failure of TP loss is understandable since it is very hard for predicting the next transition when half of the ground truth is noisy ($p_{tr} = 0.5$). On the other hand, TR loss facilitates easier learning wherein the model only needs to reconstruct past observations which are somehow already encoded in the current representation.

| Game | Nature DQN | MFEC | NEC | MBEC++ |
|---|---|---|---|---|
| Alien | 634.8 | 1717.7 | **3460.6** | 1991.2 |
| Amid | 126.8 | 370.9 | **811.3** | 369.0 |
| Assault | 1489.5 | 510.2 | 599.9 | **4981.3** |
| Asterix | 2989.1 | 1776.6 | 2480.4 | **7724.0** |
| Asteroids | 395.3 | **4706.8** | 2496.1 | 1456.2 |
| Atlantis | 14210.5 | 95499.4 | 51208.0 | 99270.0 |
| Bank Heist | 29.3 | 163.7 | 343.3 | 1126.4 |
| Battlezone | 6961.0 | 19053.6 | 13345.5 | 30004.0 |
| Beamrider | 3741.7 | 858.8 | 749.6 | **5875.2** |
| Berzerk | 484.2 | **924.2** | 852.8 | 759.2 |
| Bowling | 35.0 | 51.8 | 71.8 | **80.6** |
| Boxing | 31.3 | 10.7 | 72.8 | **95.8** |
| Breakout | 36.8 | 86.2 | 13.6 | **372.2** |
| Centipede | 4401.4 | **20608.8** | 12314.5 | 8693.8 |
| Chopper Command | 827.2 | 3075.6 | **5070.3** | 1694.0 |
| Crazy Climber | 66061.6 | 9892.2 | 34344.0 | **107740.0** |
| Defender | 2877.90 | 10052.80 | 6126.10 | **690956.0** |
| Demon Attack | 5541.9 | 1081.8 | 641.4 | 8066.4 |
| Double Dunk | -19.0 | -13.2 | **1.8** | -1.8 |
| Enduro | **364.9** | 0.0 | 1.4 | 343.7 |
| Fishing Derby | -81.6 | -90.3 | -72.2 | **17.6** |
| Freeway | 21.5 | 0.6 | 13.5 | **33.1** |
| Frostbite | 339.1 | 925.1 | **2747.4** | 1783.0 |
| Gopher | 1111.2 | 4412.6 | 2432.3 | **11386.4** |
| Gravitar | 154.7 | 1011.3 | **1257.0** | 428.0 |
| H.E.R.O. | 1050.7 | 14767.7 | **16265.3** | 12148.5 |
| Ice Hockey | -4.5 | -6.5 | -1.6 | **-1.5** |
| James Bond | 165.9 | 244.7 | 376.8 | **898.0** |
| Kangaroo | 519.6 | 2465.7 | 2489.1 | **16464.0** |
| Krull | 6015.1 | 4555.2 | 5179.2 | **9031.38** |
| Kung Fu Master | 17166.1 | 12906.5 | 30568.1 | **37100.0** |
| Montezuma's Revenge | 0.0 | **76.4** | 42.1 | 0.0 |
| Ms. Pac-Man | 1657.0 | 3802.7 | **4142.8** | 2687.2 |
| Name This Game | 6380.2 | 4845.1 | 5532.0 | **7822.8** |
| Phoenix | 5357.0 | 5334.5 | 5756.5 | **15051.8** |
| Pitfall! | **0.0** | -79.0 | **0.0** | **0.0** |
| Pong | -3.2 | -20.0 | 20.4 | **20.8** |
| Private Eye | 100.0 | **3963.8** | 162.2 | 100.0 |
| Q*bert | 2372.5 | **12500.4** | 7419.2 | 8686.0 |
| River Raid | 3144.9 | 4195.0 | 5498.1 | **10656.4** |
| Road Runner | 7285.4 | 5432.1 | 12661.4 | **55284.0** |
| Robot Tank | 14.6 | 7.3 | 11.1 | **23.9** |
| Seaquest | 618.7 | 711.6 | 1015.3 | **10460.2** |
| Skiing | -19818.0 | -15278.9 | -26340.7 | -10016.0 |
| Solaris | 1343.0 | **8717.5** | 7201.0 | 1692.0 |
| Space Invaders | 642.2 | **2027.8** | 1016.0 | 1425.6 |
| Stargunner | 604.8 | 14843.9 | 1171.4 | 49640.0 |
| Tennis | 0.0 | -23.7 | -1.8 | **18.8** |
| Time Pilot | 1952.0 | **10751.3** | 10282.7 | 6752.0 |
| Tutankham | 148.7 | 86.3 | 121.6 | 206.36 |
| Up'n Down | 18964.9 | 22320.8 | **39823.3** | 21743.2 |
| Venture | 3.8 | 0.0 | 0.0 | **1092.4** |
| Video Pinball | 14316.0 | 90507.7 | 22842.6 | **182887.9** |
| Wizard of Wor | 401.4 | **12803.1** | 8480.7 | 6252.0 |
| Yars' Revenge | 7614.1 | 5956.7 | 21490.5 | **21889.8** |
| Zaxxon | 200.3 | 6288.1 | 10082.4 | **11180.0** |

Table 3: Scores at 10 million frames.

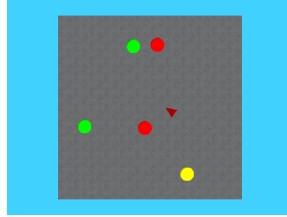 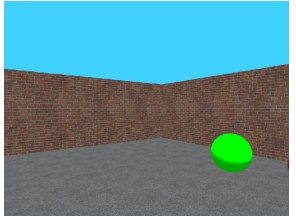 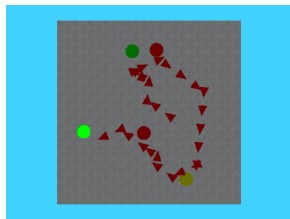

Figure 11: 3D Navigation: picking 5 objects task. Top view map (left) and frontal view-port or the observed state of the agent (middle) and a solution found by MBEC++ (right).

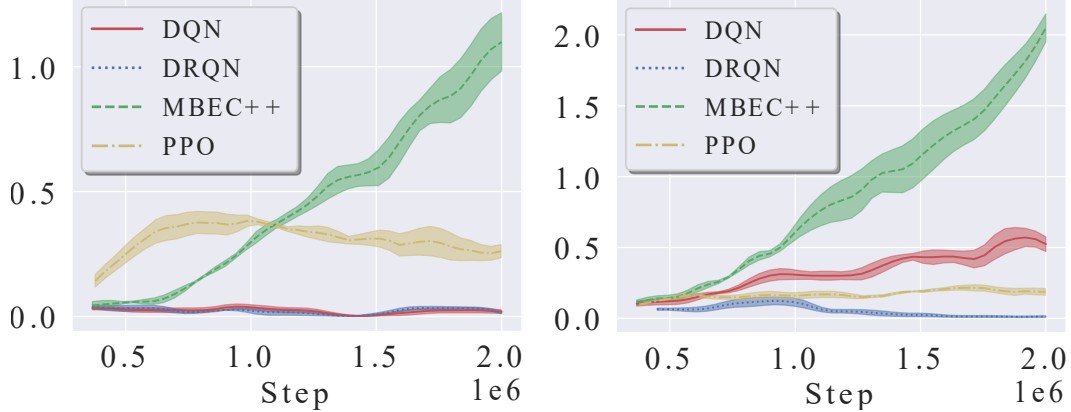

Figure 12: 3D Navigation. Average reward for picking 3 (left) and 5 (right) objects over environment steps (mean and std. over 5 runs).

## B.4 Atari 2600 task

We use Adam optimizer with a learning rate of $10^{-4}$, batch size of 32 and only train the models within 10 million frames for sample efficiency. Other implementations follows [34] (CNN architecture, exploration rate, 4-frame stacking, reward clipping, etc.). In our implementation, at timestep t, we use frames at t,t-1,t-2,t-3 and still count t as the current frame as well as the current timestep. We tune the hyper-parameters for MBEC++ using the standard validation procedure, resulting in $K = 7$, $L = 100$ and $N = 50,000$.

We also follow the training and validation procedure from [34] using a public DQN implementation[6]. The CNN architecture is a stack of 4 convolutional layers with numbers of filters, kernel sizes and strides of $[32, 64, 64, 1024], [8, 4, 3, 3]$ and $[4, 2, 1, 1]$, respectively. The Atari platform is Open AI's Atari environments[7].

In order to understand better the efficiency of MBEC++, we record and compare the learning curves of MBEC++, MBEC and DQN (our implementation using the same training procedure) in Fig. 9. We run the 3 models on 6 games (Alien, Breakout, Fishing Derby, Asterix, Gopher and Hero), and plot the average performance over 5 random seeds. We observe a common pattern for all learning curves, in which the performance gap between MBEC++ and DQN becomes clearer around 5 million steps and gets wider afterwards. We note that MBEC++ demonstrate fast learning curves for some games (e.g., Breakout and Fishing Derby) that other methods (DQN, MFEC or NEC) struggle to learn.

We realize that early stopping of training trajectory model or reward model does not affect the performance much as the quality of trajectory representations and reward prediction is acceptable at about 5 millions steps (see Fig. 10). Early stopping further accelerates the running speed of MBEC++ and also helps stabilize the learning of the Q-Networks.

---

[6] https://github.com/Kaixhin/Rainbow
[7] https://gym.openai.com/envs/#atari

| Model | Alien | Asterix | Breakout | Fishing Derby | Gopher | Hero |
|-------|-------|---------|----------|---------------|--------|------|
| Dreamer-v2 | **2950.1** | 3100.8 | 57.0 | -13.6 | **16002.8** | **13552.9** |
| Our MBEC++ | 1991.2 | **7724.0** | **372.2** | **17.6** | 11386.4 | 12148.5 |

Table 4: Dreamver-v2 vs MBEC++ on 6 Atari games at 10M frames. We report the best results of the models after three runs.

| Model | Alien | Asterix | Breakout | Fishing Derby | Gopher | Hero |
|-------|-------|---------|----------|---------------|--------|------|
| SIMPLE♣ | **378.3 ± 85.5** | 668.0 ± 294.1 | 6.1 ± 2.8 | -94.5 ± 3.0 | **510.2 ± 158.4** | 621.5 ± 1281.3 |
| Our MBEC++ | 340.5 ± 39.7 | **810.1 ± 42.4** | **11.9 ± 2.0** | **-81.5 ± 2.3** | 459.2 ± 60.4 | **1992.8 ± 1171.9** |

Table 5: SIMPLE vs MBEC++ on 6 Atari games at 400K frames. Mean and std over 5 runs. ♣ is from [23].

Table 3 reports the final testing score of MBEC++ and other baselines for all Atari games. We note that we only conducted five runs for the Atari games mentioned in Fig. 9. For the remaining games, our limited compute budget did not allow us to perform multiple runs, and thus, we only ran once. We store the best MBEC++ models based on validation score for each game during training and test them for 100 episodes. Other baselines' numbers are reported from [36]. Compared to other baselines, MBEC++ is the winner on the leaderboard for about half of the games.

Notably, our episodic memory is much smaller than that of others. For example, NEC and EMDQN maintain 5 millions slots per action (there are total 18 actions). Meanwhile, our best number of memory slots $N$ is only 50,000.

## B.5   3D navigation task

In this task, the agent's goal is to pick objects randomly located in a big room[8]. There are 5 possible actions (moving directions and object interaction) and the number of objects is customizable. We train MBEC++, DQN and DRQN using the same training procedure and CNN for state encoding as in the Atari task. Except for DQN, other baselines (DRQN and PPO) uses LSTM to encode the state of the environment. We also stack 4 consecutive frames to help the models (especially DQN) cope with the environment's limited observation. For PPO, we tuned the clipping threshold {0.2, 0.5, 0.8} and reported the best result (0.2).

Fig. 11 illustrates one sample of the environment map and a solution found by MBEC++. The best hyper-parameters for MBEC++ are $K = 15$, $L = 20$ and $N = 10000$.

## B.6   Ablation study

**Classical control**   We tune MBEC++ with Noisy Transition Mountain Car problem using range $K \in \{1, 5, 15\}$, $L = \{1, 10, 50\}$, $N = \{500, 3000, 30000\}$. We use the best found hyper-parameters ($K = 15$, $L = 10$, $N = 3000$) for all 9 problems. The learning curves of MBEC++ in Noisy Transition Mountain Car are visualized in Fig. 5 (the first 3 plots).

We also conduct an ablation study on MBEC++: (i) without TR loss, (ii) with TP loss instead (iii), without multiple write ($K_w = 1$) and (iv) without memory refining. The result demonstrates that ablating any components reduces the performance of MBEC++ significantly (see Fig. 5 (the final plot)).

**Dynamic consolidation**   We compare our dynamic consolidation with traditional fixed combinations in both simple and complex environments. Fixed combination baselines use fixed $\beta$ in Eq. 7, resulting in

$$Q\left(s_t, a_t\right) = Q_{MBEC}\left(s_t, a_t\right)\beta + Q_\theta\left(s_t, a_t\right)$$

In CartPole (Gaussian noisy reward), all fixed combinations achieve moderate results, yet fails to solve the task after 10,000 training steps. Dynamic consolidation learning to generate dynamic $\beta$, in contrast, completely solves the task (see Fig. 13 (a)).

---

[8] `https://github.com/maximecb/gym-miniworld`

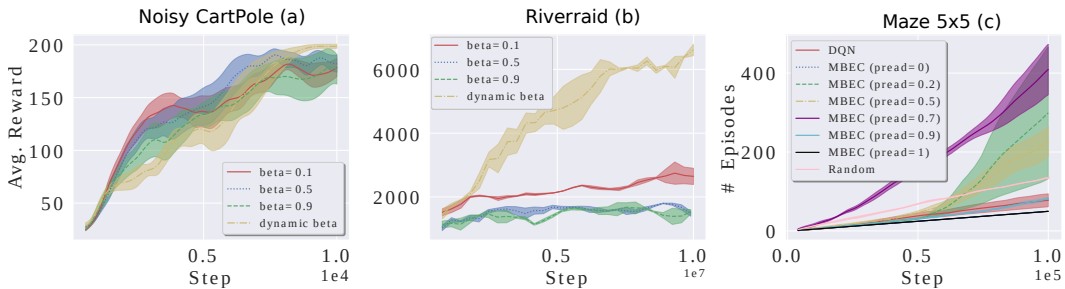

Figure 13: Ablation studies on dynamic consolidation (a,b) and $p_{read}$ (c). All curves are reported with mean and std. over 5 runs.

In Atari game's Riverrraid–a more complex environment, the performance gap between dynamic $\beta$ and fixed $\beta$ becomes clearer. Average reward of dynamic consolidation reaches nearly 7,000 while the best fixed combination's ($\beta = 0.1$) is less than 3,000 (see Fig. 13 (b)).

**Tuning $p_{read}$** Besides modified modes introduced in the main manuscript, we investigate MBEC with different $p_{read}$ and DQN in a bigger maze ($5 \times 5$) for the original setting. As shown in Fig. 13 (c), many MBEC variants successfully learn the task, significantly better in the number of completed episodes compared to random and DQN agents. We find that when $p_{read} = 1$ (only average reading), the performance of MBEC is negatively affected, which indicates that the role of max reading is important. Similar situation is found for for $p_{read} = 0$ (only max reading). Among all variants, $p_{read} = 0.7$ shows stable and fastest learning. Hence, we set $p_{read} = 0.7$ for all other experiments in this paper.