# OpenReview forum: "Model-Based Episodic Memory Induces Dynamic Hybrid Controls"
_NeurIPS.cc/2021/Conference — NeurIPS 2021 Poster_

### Official Review · Reviewer_BrPf · 2021-07-09

**Rating:** 7
**Confidence:** 4

**Summary:**

This paper introduces a novel episodic memory mechanism for deep RL. The episodic memory module learns an embedding space for trajectories and stores associated Monte Carlo value estimates in a dictionary. While acting in the world, the agent is then able to compare the embedding of its current trajectory to its memory and form a value estimate which can then be used on its own to guide action selection or in conjunction with model-free methods. Experiments on a variety of domains demonstrate the performance of this framework.

**Limitations And Societal Impact:**

See the main review for a discussion of limitations and suggestions for improvement. I do not believe there are obvious societal impacts for this work.

**Main Review:**

Originality: The primary novel contributions of the model are the learned trajectory embedding and the memory writing/updating mechanisms. These additions are interesting and, compared to Blundell et al. (2016), seem to better reflect the functioning of episodic memory in the brain—we store representations of sequences of events/contextualized experiences, not isolated states or state-action pairs—although of course any mapping to neural architecture is loose at best. The addition of the dynamic weighting between memory-based and parametric values estimates is also novel, though similar ideas are common in the literature. However, this work is strongly related to previous work on episodic memory in RL (particularly MFEC), and the novel contributions should be made more clear. MFEC is not even mentioned in Section 2.2, and MBEC is portrayed as the first deep RL method to explicitly attempt to model episodic memory, which is not the case.

Quality & Significance: In general, the various ablations and experiments support the effectiveness of the various components of the model. However, some concerns remain (as the authors note, the number of moving parts in the setup makes it hard to verify the effectiveness of any one component in isolation). First, I wonder why MBEC wasn’t also applied to the stochastic, Atari, and POMDP settings—MBEC vs. MFEC seems a more fair comparison than MBEC++ vs. MFEC. Relatedly, it would be more fair to compare MBEC against other model-based approaches to deep RL (e.g., Dreamer-v2 or PlaNet), rather than model-free methods like DQN and MFEC. Finally, in order to more properly understand the usefulness of the trajectory model, it would be helpful to provide more analysis as to the reasons behind the poor performance of MFEC and/or DQN in certain experiments (e.g., the dynamic maze). Would any model-based method solve these issues? Finally, in Figure 2c, why do the steps start at 80k? What happens before that point?

Clarity: The paper is generally easy to follow given the number of different components in the model. However, I did find the righthand panel in Figure 4 difficult to parse. I assume that the main takeaway from parts (b-d) is that the trajectories remain coherent despite the noisy time steps, but it’s unclear what the axes represent in these cases/how to interpret them. Labels and additional explanation would be helpful.

In light of the above factors, I tend to give this paper a borderline accept. I think the novel components of the model are interesting and the motivating questions are important and interesting for RL. However, it would be nice to see a more detailed contextualization of MBEC in reference to related work, as well as a measure of its performance/qualities vs. standard model-based approaches to deep RL. If the concerns above are addressed, I would be willing to increase my score.

**Time Spent Reviewing:**

6 hours

---

> ### Author Response · Authors · 2021-08-10
> **Reply to Reviewer BrPf**
>
> We thank the reviewer for your insightful and constructive comments. We address your questions in the following.
>
> $\textbf{"However, this work is strongly related ..."}$ We described our novel contributions in L36-54 (motivation and how they differ from other episodic controls).
>
> $\textbf{"MFEC is not even mentioned ..."}$ Section 2.2 is about other memory models in RL. A review of various episodic memories including MFEC is introduced early in the introduction (L18-27). Due to page constrain, we did not repeat it in the related work. In the revision, we will try to add more content to the related work.
>
> $\textbf{"First, I wonder ..."}$ We could have included the results for MBEC (worse than MBEC++ and on par with MFEC on Classical Control task). Yet, we did not have enough computation resources to carefully tune both MBEC and MBEC++. So we kept using and tuning only MBEC++ on other tasks. We agree that it is fairer to compare MBEC with MFEC (and we did it in the maze task). Note that for other tasks, we compared MBEC++ with various baselines, not only MFEC. MFEC is not a strong baseline for these tasks.
>
> $\textbf{"Relatedly, it would be ..."}$ We want to improve episodic control (EC) so we compare with SOTA EC, not model-based RL methods. Also, the Atari setting of Dreamer-v2 (200M frames) and SIMPLE (low-regime up to 4M frames) are different from the common episodic control setting (10M), which makes it harder to compare (PlaNet is for Mujoco). If we just look at the reported numbers, ours is far better than SIMPLE and if we check the learning curves of Dreamer-v2 at 10M frames, ours also shows competitive results. That comparison may not be totally fair, yet it gives a rough view of how our approach performs w.r.t recent model-based RL.
>
> $\textbf{"Finally, in order to ..."}$ We hypothesize that MFEC and DQN only capture states, which can be ambiguous in this nonstationary setting (as explained in L221-222). Capturing trajectories, on the other hand, is less ambiguous (it is unlikely that there exist similar trajectories in different mazes). We examined a simple model-based approach (TP loss) and found it performed much better than MFEC and DQN though not as good as our TR loss (Fig. 2c).
>
> $\textbf{Fig. 2c}$ It is an error. The x-axis should start from timestep 0. Thank you for pointing it out.
>
> $\textbf{Clarity}$ The axes are the dimension of the trajectory vectors $\vec{\tau}$ (only 2 in this case). We will add labels and explanations as suggested.

---

> > ### Comment · Reviewer_BrPf · 2021-08-13
> > **Response**
> >
> > Thank you for your detailed response. To address your points in order:
> >
> > - I appreciate this. Perhaps a sentence or two simply clarifying the key novel elements of the model. As it stands, there are a lot of moving parts, and lines 36-54 currently read more like a list of everything that's done in the paper, rather than focusing only the novel components, so it could be slightly confusing.
> >
> > - Thank you, and I appreciate that the page constraint makes things difficult.
> >
> > - If it would be possible to have this comparison in the camera-ready version of the paper, that would be nice to see.
> >
> > - I don't really subscribe to the idea that just because the focus is on the EC aspect of the model, it completely absolves one from also situating the contributions within the model-based RL literature. The approach is fundamentally *both* EC and model-based. If it's not possible to compare performance, I think that there should at least be a fairly thorough discussion (even just in the appendix) comparing similarities/differences/expected performance between MBEC and other MBRL methods. The MFEC paper compares against several flavors of DQN, so if there's an "upgrade" to model-based learning, I still feel there should be some comparison to model-based methods.
> >
> > - Re: the last two points: Thanks!
> >
> > Given the above, I'll keep my current score for now, but if there's a commitment to addressing in some substantial manner the concerns I've listed above in the final version of the paper, I'd be happy to consider raising it.

---

> > > ### Author Response · Authors · 2021-08-16
> > > **Ideas for improvement**
> > >
> > > We thank the reviewer for your insightful and constructive feedback. We appreciate your understanding of the page limit. In the final revision, we will be addressing your remaining concerns as follows.
> > >
> > > $\textbf{Novel contribution}$. We will add few sentences to clarify our novel contributions in the introduction, which are mainly about:
> > >
> > > - Our episodic memory that stores representation of the trajectory instead of states and actions
> > >
> > > - Our memory-based value estimation that leverages memory lookup for planning
> > >
> > > - Our dynamic consolidation that adaptively combines episodic and parametric values
> > >
> > > $\textbf{More experiments with MBEC}$. We already had MBEC results for Classical control tasks and will be adding them to Fig. 3. We are going to run additional Atari experiments with MBEC (for at least six games and the results will be added to Fig.  9).
> > >
> > > $\textbf{Comparison with model-based RL}$. Thank you for your ideas. We will be adding a subsection on model-based RL in the related work and appendix, including the following points:
> > >
> > > - Reference to recent model-based RL (Dreamer, PlaNet, SIMPLE)
> > >
> > > - Discussion on the similarities and differences between ours and others (the way the models are constructed and employed)
> > >
> > > - Discussion on the expected performance (similar to the fourth point in our previous reply)
> > >
> > > If you still have any concerns, feel free to give further comments. Thank you.

---

> > > > ### Comment · Reviewer_BrPf · 2021-08-30
> > > > **Response**
> > > >
> > > > Thank you very much for your detailed response and the new model-based results. I'm updating my score to a 7 accordingly.

---

> > > > > ### Author Response · Authors · 2021-08-31
> > > > > **Thank you**
> > > > >
> > > > > Thank you for increasing your score.

---

### Official Review · Reviewer_G6K7 · 2021-07-15

**Rating:** 6
**Confidence:** 3

**Summary:**

This paper proposes a complimentary learning system combining a non-parametric episodic controller with a value function learnt by DQN. The method is extensively evaluated on a wide range of tasks from gridworld and classic continuous control problems to all Atari games and a 3D partially observable navigation task.

**Limitations And Societal Impact:**

There is only a very short discussion in the conclusion of the methods limitations and societal impact. It would improve the paper to also discuss environments or use cases where this approach would not be applicable, would be less effective or could have societal impact (positive and negative). Doing so gives readers greater insight into how the method works and highlights opportunities for future work. I am keen to hear a more detailed consideration of these issues in the author's reply.

**Main Review:**

The method appears novel by complexity (I doubt this precise combination of losses and network architecture has been published before) but demonstrates a sufficient performance gain in a wide range of suitable environments to be of wide interest. It is compared to a good range of algorithms, but the significance of the performance gains could be clearer if there was justification for why these specific algorithms were chosen as baselines.

Similarly other choices in the methods design are presented without justification. For example, on line 137 "the reading rules are simply selected randomly" - what informed this choice? were experiments run with alternative reading rules?

Figure 1 is a great resource to improve the clarity of the proposed method, but appears to be missing key details. In particular the reward model's loss (Equation 5) is not included and the legend could include more details explicitly (e.g. the X and + to the right of Q_MBEC). The coloring of the figure also appears inconsistent (purple is a writing step but also Q_MBEC?) and may not be an accessible way to convey information to all readers. I encourage the authors to revise this figure, making full use of the width of the page (as there is a significant amount of unused space either side) and linking the figure to the sections of the text that elaborate on specific parts of the method.

The empirical evaluation of the method is extensive and informative. However, I would like to hear during the discussion period: (1) why figure 2c has no variance illustrated; and (2) why efficiency improvement was chosen as the metric to plot for this experiment? Perhaps I am missing a key insight here that would not be present in the more typical reward curves shown for other experiments?

In the appendix, details are included for how some methods were tuned but it is unclear if equal computation was given to tuning all methods given the differences in hyperparameters. Particularly for PPO, as no mention of tuning the algorithm to this environment is noted in lines 539-542. Please clarify how balanced the allocation of compute resources to tuning each algorithm was. This significantly affects the fairness of the study, and my interpretation of the significance of the results shown.

Minor Comments (no reply needed):
- On line 103 the sentence "for a considering action" does not parse. Should this perhaps be "for any action considered "?

- Several key details (e.g. explanation of the TR loss) and contributions (e.g. empirical results for the 3D POMDP) are not included in the main paper. This appears to be due to the quantity of material (particularly the large number of empirical evaluations). In my opinion, the clarity of the paper would be improved if more of the key details of the method were in the main paper and more of the empirical results were moved to the appendix. In particular the sections on the POMDP and Atari games could be moved to free up almost an additional page of explanation of the method. However, I appreciate the preference for arrangement of material is subjective and so this does not significantly affect my review score.


**Time Spent Reviewing:**

5

---

> ### Author Response · Authors · 2021-08-10
> **Reply to  Reviewer G6K7**
>
> We thank the reviewer for your valuable feedback. We address your concerns in the following.
>
> $\textbf{L137}$ Taking the sum/mean will make the combined rule behave like the average reading rule. To keep the property of each rule, using them separately at different frequency is the best way. This motivates us to select the rule randomly. We had an ablation study to compare our reading rule with others (Appendix L659-666)
>
> $\textbf{Figure 1}$ Thank you for your suggestions. We will revise it accordingly.
>
> $\textbf{Figure 2c}$ The variance was too small (almost invisible in the plot), so we did not visualize it. Here, we want to show a different perspective. Since in this maze task, plotting the rewards does not illustrate clearly the superiority of good agents. Random agents can also earn many rewards as the maze is small. Thus, it is better to compare the relative improvement over a random agent.
>
> $\textbf{"In the appendix, details are ..."}$ We reduced the effort of tuning MBEC++ (Appendix L502). In practice, we mainly tuned K. For PPO, we tuned the clipping threshold {0.2, 0.5, 0.8} and reported the best result (0.2). We will add this information in the next revision. For other baselines, we extensively tuned them as stated in Appendix B1.
>
> $\textbf{Minor Comments}$ Thank you for your comments.

---

> > ### Comment · Reviewer_G6K7 · 2021-08-16
> > **Outstanding Questions**
> >
> > Thank you for this clear response to many of the queries raised in this review. However, one question I raised that could significantly affect my rating of this paper remains unanswered. In hindsight I realize I could have made it clearer that I was seeking a response to this during the rebuttal period. So, if the authors are still available, I invite them to respond to this query in the main review of:
> >
> > **-  Why were these specific algorithms chosen as baselines?**
> >
> > And if their time permits, to further discuss environments or use cases where this approach would not be applicable, would be less effective or could have societal impact (positive and negative).

---

> > > ### Author Response · Authors · 2021-08-18
> > > **Discussion on baselines and use cases**
> > >
> > > Thank you for your questions. We are sorry that we missed these in our previous reply. We answer your remaining questions in the following.
> > >
> > > $\textbf{ "Why were these specific algorithms chosen as baselines?"}$ First, we note that we want to improve episodic control (EC) so many baselines in our paper fall into EC methods. DQN is the basic method so it appears on all tasks as the standard baseline. The motivations for choosing other baselines depend on tasks.
> > >
> > > In the simple maze tasks, we aim to highlight the difference between our memory MBEC (trajectory, model-based) and the classical MFEC (state-action, model-free). The goal is to measure memory capability alone, so we did not include hybrid memory models using TD learning.
> > >
> > > In the classical control tasks, we include stronger hybrid memories such as MBEC++ and NEC. NEC is chosen since its public code is available on PyTorch and is easy to adapt to our customized tasks.
> > >
> > > In the Atari task, we follow the common practice of using a standard benchmark. Here, we include all prominent EC baselines whose performances are reported in previous publications (see Zhu et al., Episodic reinforcement learning with associative memory. ICLR2019 for the list of recent methods).
> > >
> > > In the navigation task, we use DRQN  because it is designed for POMDP. PPO is the default baseline for this environment. Both DRQN and PPO use LSTM encoders, which can be fair competitors to our LSTM trajectory model.
> > >
> > > $\textbf{"And if their time permits ..."}$. Thank you for your suggestion.
> > > In the revision, we will add more discussion relating to this point as follows,
> > > - There are situations where episodic memory, in general, is not required (simple environments where the reward is dense and the episode is short). Our method is expected not to show an advantage in these cases.
> > > - Another inapplicable case is environments with continuous action space. Currently, our planning mechanism simply iterates over discrete actions. Extending to continuous action space is an interesting future work (mentioned in Discussion). Or in highly non-stationary environments, our method may not work really well. As seen in Fig. 2c, in the difficult setting, we improve 20% over the random agent and still have room for further improvement.
> > > - Regarding positive societal impact, our method enables sample efficiency in training RL agents, thereby reducing compute cost and energy for training (better for the environment). Our method improves the performance in various RL tasks, and thus opens the chance for creating better autonomous systems that work flexibly across sectors (robotics, manufacturing, logistics, and decision support systems)
> > > - Regarding negative societal impact, although we do not think there are immediate bad consequences, we are aware of potential problems. First, our method does not guarantee safe exploration during training. If learning happens in a real-world setting (e.g. self-driving car), the agent can make unsafe exploration (e.g. causing accidents). Second, we acknowledge that our method, like many other Machine Learning algorithms, can be misused in unethical or malicious activities. For example, due to the sample efficiency, small groups can still utilize our method to train autonomous systems for cyber attacks. That said, we will do our best to stop systematic violations of ethics in AI from our end.

---

> > > > ### Comment · Reviewer_G6K7 · 2021-08-18
> > > > **Re: Baselines**
> > > >
> > > > Thank you for the detailed response, this helps me appreciate the decisions made regarding the baselines already included within the paper. However, I am less convinced by the responses to the other reviewers regarding not comparing directly against model-based baselines (e.g. Dreamer-v2 and SIMPLE). I agree with reviewer BrPf that there should be both EC and model-based baselines (or ablations) as the proposed approach is fundamentally both. The proposed comparison to the previously published results under significantly longer (Dreamer-v2) and shorter (SIMPLE) training times provides some preliminary insight but it would be significantly better to train these approaches under the same 10M settings. As the codebase for both are open source and already integrated with an environment included in this study this seems a relatively straight forward task to significantly improve the paper.
> > > >
> > > > https://github.com/danijar/dreamerv2
> > > >
> > > > https://github.com/tensorflow/tensor2tensor/tree/master/tensor2tensor/rl

---

> > > > > ### Author Response · Authors · 2021-08-23
> > > > > **Baseline results**
> > > > >
> > > > > Thank you for sharing the link to the code. We have used them and showed initial results in our common comment above. We hope that this somehow help clarify your concerns regarding the baselines.

---

> > > > > > ### Comment · Reviewer_G6K7 · 2021-09-13
> > > > > > **Thank you**
> > > > > >
> > > > > > Thank you for your detailed responses and engagement throughout the discussion period. I think including the additional results you shared here and the justification given for the chosen baselines in response to my review and further questioning will improve the paper. Ideally of course the model based baselines would also be ran on all Atari games, but I appreciate the compute and time constraints to run those experiments at this time. Therefore I remain in support of accepting this paper.

---

### Official Review · Reviewer_grEv · 2021-07-18

**Rating:** 6
**Confidence:** 5

**Summary:**

The authors introduce a novel hybrid model, inspired by the complementary learning systems hypothesis, that claims to improve performance on a wide range of tasks. This model includes a standard parametric value function (e.g. DQN) and a episodic buffer relating learned trajectory representations to (almost) monte-carlo value estimates. I say almost because while this is true at initialization, a similarity-based heuristic is used to update stored values such that similar vectors end up with similar values.

The trajectory representations are learned via an auto-regressive prediction process that (I believe) utilizes a pretrained network to avoid reconstruction of high-dimensional observations.

The two value estimates (parametric and non-parametric) are combined via a weighting function which attempts to minimize the temporal difference error of the aggregate value function.


**Limitations And Societal Impact:**

yes

**Main Review:**

This work is full (perhaps too full) of unique modifications to the basic idea of combining parametric and non-parametric value estimates for improved data efficiency, and the results do seem quite impressive. But a proper evaluation of this algorithm is greatly hindered by persistent issues involving clarity and a lack of sufficient detail.

The most glaring example is the issue of using a pretrained network to yield targets for the TR/TP losses. What loss is this network trained on? What is its architecture? Is it trained in tandem with the other components, or is there a separate pre-training phase? I poured over both the paper and appendix and can't find any answers. And these answers should greatly impact the overall merit of this paper. If everything is trained in tandem, then the empirical results are very solid. But if a long-pretraining period and/or highly curated data are used, then all of the experimental results are comparing apples to oranges.

The TR/TP distinction is confusing. It's mentioned in the main text, but only explained in the appendix. The explanation is that TP uses time t to predict time t+1, whereas TR uses t-1 to predict time t (line 561). But for the sorts of environments under consideration, I can't see how this distinction matters. On line 118, it is mentioned that a "noisy" version of the state-action tuple is used in the TR loss, perhaps this isn't the case for the TP loss? Relatedly, how is this "noisy" version constructed?

The weighting function is another place where more explanation is needed. As mentioned in my summary, I believe its trained to minimize the TD error of the aggregate value function (Equation 8). But why is this a reasonable thing to do, considering your changing the mixture weight rather than the value estimates directly? For example, the parametric values will initially collapse to zero on many environments with sparse rewards. In which case, it'd be optimal to weigh this value highly since it minimizes TD error, but this feels like exactly the case where you'd want non-parametric estimates to be up-weighted.

I like the explicit list of open problems this works resolves (line 28), but the proposed fix for the problem of needing deterministic environments doesn't make sense (line 36). MFEC requires deterministic environments because it behaves based on sampled experiences with the highest return. Thus, a single winning lottery ticket would forever force MFEC to buy more tickets, since it only considers the best return rather than the expected. MBEC's reconstruction loss doesn't impact this at all. Rather, MBEC is likely more robust to stochasticity due to it averaging over nearest neighbors rather that simply taking the max. Though it still does that sometimes (Line 133), and to the extent that it does, it also is biased towards deterministic environments.

This is a bit of a quibble, but the psychology angle hurts clarity and doesn't sufficiently connect to the literature enough to make up for it. Parametric vs non-parametric is a lot more explicit than episodic vs semantic. Semantic and habitual also seem to be used synonymously, which isn't the case in psychology nor machine learning. Learning a forward model makes sense on its own, and relating it to a serial recall test doesn't better motivate it (line 118). I'd suggest all text relates the method to complementary learning systems hypothesis be relegated to the discussion section.

The experimental methods for Atari are quite lacking. Rather than report any notion of spread, only the best seeds appear to be reported (line 626), and it is unclear how many seeds were run.

The ablations that are there are appreciated, but this paper adds many more distinction components that aren't ablated, e.g. is switching between two reading rules really necessary? I believe the ablation in the appendix (Figure 13) is showing the effect of dynamic weighting between the two value functions (again, this is unclear due to lack of supporting text). If so, there doesn't seem to be any support for it outside of a single Atari game. If an ablation shows a component isn't needed, eliminate it from the method.

EDIT: several of my points were misunderstandings, and my others were largely addressed through the additional baselines and clarifications. I'm conditionally voting to accept under the assumption that the final draft is explicit about experimental seeds and other sources of confusion.


**Time Spent Reviewing:**

4

---

> ### Author Response · Authors · 2021-08-10
> **Reply to Reviewer grEv**
>
> We thank the reviewer for your valuable feedback. We address your concerns in the following.
>
> $\textbf{"The most glaring example is the ..."}$ There seems to be a misunderstanding. We did $\textit{not}$ use any pretrain network to construct TR/TP loss. In Eq. 1, the target is the observations from the environment (s, a) and the networks are $\omega$ and $\phi$. The networks are trained to reconstruct the data from the environment so there is no need for a pretrained target network. We note that $\omega$ and $\phi$ are jointly trained with other components (Algo. 1 line 13)
>
> $\textbf{"The TR/TP distinction ..."}$ TP loss is the common transition prediction often used in model-based RL. TR loss is very different from TP loss. Reconstructing the past is easier than predicting the future since the information of past timesteps {t-k, t-k+1} are already embedded in the hidden state at timestep t of the LSTM modelling the trajectory, which is always possible for reconstruction somehow. On the other hand, sometimes it is impossible to predict the future (as in noisy and stochastic environments). From a theoretical point of view, TR loss is also more beneficial than TP loss under our setting (see Appendix 1).
>
> $\textbf{L118}$ As stated in Appendix L508, we add Gaussian noise to the observations. We applied noise to both TR and TP loss. We will revise Eq. 10 to make this point clearer.
>
> $\textbf{"The weighting function ... "}$ Adding $Q_{MBEC}$ to Eq. 8 injects the episodic value to train the parametric model. $Q_{MBEC}$ in the target provides better target value estimation in the early phase of learning when the parametric model does not learn well (as your example indicates).  We will make this point clearer in the next revision.
>
> $\textbf{"I like the explicit list ..."}$ The loss contributes to tackling problem (a) because it helps learn a noise-tolerance representation of the trajectory (see an example in Fig. 4 right). The main point is if the representation is good (also the key in the memory), even with stochastic or noisy observations, we can still choose a precise set of neighbours for reference. Without that, the average rule cannot work (averaging values of wrong neighbours is useless). That said, we agree that the averaging rule plays an important role in tackling the problem (a) and will revise L28 to reflect this point.
>
> $\textbf{"This is a bit of a quibble ..."}$ We will try to revise according to your suggestions. There are few points we would like to correct. We do not mean semantic and habitual are synonymous. As stated in L13, habitual learning is cached control or TD learning in model-free methods. We will revise L24 to void confusion. L118, we use serial recall test to motivate the formulation of the trajectorial loss, not the forward model.
>
> $\textbf{"The experimental methods ..."}$ As stated in Appendix L617, we ran over 5 random seeds and reported the mean and std. in Appendix Fig. 9 for several games.
>
> $\textbf{"The ablations that are ..."}$ Appendix B6 listed more details of our ablation study. Particularly, two reading rule ablation was mentioned in Appendix L659-666. Supporting texts for dynamic weighting was in Appendix L650-658 (2 environments were tested, not only Atari game). Our results show that all the proposed components are necessary.

---

> > ### Comment · Reviewer_grEv · 2021-08-16
> > **Experimental methods question**
> >
> > Thanks for your informative rebuttal!
> >
> > I apologize for missing the reading rule ablation, and between that and the pretraining/joint training misunderstanding, I'm feeling a lot better about the paper.
> >
> > That said, I wanted to followup on the experimental methods.
> >
> > " As stated in Appendix L617, we ran over 5 random seeds and reported the mean and std. in Appendix Fig. 9 for several games."
> >
> > Should I be interpreting this to mean that *all* results involved 5 seeds, or just the experiment shown in Fig. 9? If its the former, then could spread information be added to all results e.g. Table 3?

---

> > > ### Author Response · Authors · 2021-08-18
> > > **Experiment details**
> > >
> > > Thank you for your question. We are sorry that we could not make it clearer in our previous reply about Atari experiments. We will provide a detailed explanation below.
> > >
> > > Here, we only conducted five runs for the Atari games mentioned in Fig. 9.  For the remaining games, our limited compute budget did not allow us to perform multiple runs, and thus, we only ran once.  We stored the best model during training and used it to report the test performance in Table 3 (see more in Appendix L625-627).
> > >
> > > Although we did not have a chance to report spread information for all games, we realize that for the games in Fig. 9, the variance during training is not big and the difference in test performance of the best model from each run is very small. With our current resources, it will take a long time to get results with variance for all games, but that we will try our best to get more runs and add to Fig. 9 in the revision.

---

> > > > ### Comment · Reviewer_grEv · 2021-08-23
> > > > **Benefit of the doubt**
> > > >
> > > > I sympathise with your computational limitations. The work would be stronger with more than one seed, but you have shown results for multiple environments and added new baselines. And you addressed all of my other concerns quite well, so if you promise to make your exact experimental method more explicit (e.g. one seed for these games, best model, etc) I'll go ahead and change my vote to accept (score 4 --> 6).

---

> > > > > ### Author Response · Authors · 2021-08-25
> > > > > **Thank you**
> > > > >
> > > > > Thank you for increasing your score. We will add the the experimental information on seed and best run (as explained in the previous reply) to the revised manuscript.

---

### Official Review · Reviewer_trSB · 2021-07-19

**Rating:** 7
**Confidence:** 3

**Summary:**

The authors propose a reinforcement learning architecture that combines a traditional model-free/habitual component (DQN) with a fast episodic controller, flexibly trading off between the two forms of value estimation over the course of training. The episodic controller generates low-dimensional keys from state-action pairs by training to predict past observations along the agent's trajectory; these keys are matched with returns as values to generate a lookup memory for past returns from the environment. The memory is continually updated to reduce TD error. The authors illustrate strong performance across a variety of tasks (classical control, MazeWorld, medium-data Atari, 3D Navigation) and consider a number of ablations.

*****
UPDATE: After discussions with the authors about addressing some clarity issues and the inclusion of results against model-based baselines Dreamer-v2 and SIMPLE, I'm raising my final score from a 6 to a 7.
*****

**Limitations And Societal Impact:**

No issues

**Main Review:**

**Quality:**

The idea of flexibly integrating multiple learning systems is compelling from a theoretical/neuroscience standpoint. The performance is strong relative to baselines, and the appendix analyses contribute to the theoretical underpinning the various components.

My biggest difficulty with the paper is that the model is quite complicated, making it difficult to understand how all of the different components interact at once. Closely related, I think there are some clarity issues in how the model is presented (more details below).

A few additional questions about the model design:
- In Eq. 7, is any gradient propagated back into Q_{MBEC} and used to train r_{\phi}?
- If not, why train against Q(s_t,a_t) and not Q_\theta(s_t,a_t) directly in Eq. 8? In this case, also multiplying Q_\{theta}(s_t,a_t) by (1-f_{\Beta}) in Eq. 7. It seems more effective for the parametric value function to simply estimate the Q-value rather than also attempt to correct for the MBEC contribution on a step-by-step basis.
- I don't fully understand the mechanics of the memory update. In Algorithm 1 Line 9, it appears to take place online on a per-step basis; but the majority of the time, that memory will not exist yet (since it is written at the end of the episode). In fact it appears as though the "update" and "write" operations should be named in reverse, since the memory is first added to M on Line 9 (bootstrapped from the estimated TD error) and the same memory is later updated on Line 16 (using the actual returns from the current trajectory).
- Is the primary benefit of the trajectorial loss to better differentiate in partially observable environments (using past experiences), or to better approximate future on-policy behaviour (and therefore better approximate future value)? The motivation primarily focuses on the former, but it seems like the latter is as or more important for ensuring that the EC updates are valid.

**Significance:**

The authors present results in a large number of domains, suggesting that the proposed model is robust and performant across contexts compared to DQN and existing episodic controllers. Missing from the baselines are recent model-based methods (e.g. SiMPLe) which typically perform very well in the low-data domain (e.g. 100k Atari). I suspect these methods outperform the proposed model, particularly in sample efficiency. While I think the paper is meaningful for pushing the state of the art using episodic control, it's important to at least discuss comparisons against recent model-based methods, particularly when the approach is presented as a "model-based" episodic controller.

**Originality:**

The model expands on recent episodic control algorithms, particularly those that attempt to combine it with model-free methods. Like NEC, the model employs a differentiable lookup table; like EMDQN, EVA and NEC2DQN, it combines this existing algorithm with a DQN to employ multiple learning systems. MBEC uses a dynamic and differentiable tradeoff between the two learning styles, unlike the existing algorithms; NEC2DQN employs a simple decay to trade-off between them and EMDQN proposes a dynamic gating to explore in future work. Probably the most original component of the present work is the trajectorial loss function that works to better optimize the distance between points in space on the basis of shared past and future (on-policy) transitions.

**Clarity:**

Overall, I think that Section 3 might be easier to parse if you start with Eqs. 7 and 8 and work backward through the trajectory representation and the memory systems in MBEC. This way, the role of each component would be motivated first before explaining how it works.

It took me quite a while to fully understand what was going on with the trajectorial loss, particularly that it tries to reconstruct any point along the trajectory given preceding s-a pair as a cue. I think this would really benefit from a small diagram.

Specific points:
- Line 29: I think that the broader point here is about partially observable environments, of which noisy and stochastic are examples; an environment can be fully deterministic and non-noisy and still cause problems for MFEC because the state can't be determined from the image.
- Line 118 and the Appendix mention _noisy_ versions of state-action pairs. Is this injected noise? What does it mean?
- If possible, can you more clearly delineate the replay memory updates in Algorithm 1, maybe with another indented block and "for t' sampled from D do..."
- Line 190: What is the feature vector of the image state and how does it differ from the image itself?
- Line 236: State the value of p_{re} here if possible.
- Table 1 caption: "The exact numbers are not reported": what does this mean? In the original paper?
- The authors of the algorithms listed as "EEVA" and "ERLAM" in Table 1 seem to refer to their own algorithms as EVA and NEC2DQN respectively; it would probably be clearer to list them as such.

**Time Spent Reviewing:**

8

---

> ### Author Response · Authors · 2021-08-10
> **Reply to Reviewer trSB**
>
> We thank the reviewer for your insightful and constructive comments. We address your questions in the following.
>
> $\textbf{Eq. 7}$ No, the gradient is not propagated back via Q_{MBEC}
>
> $\textbf{Eq. 8}$ needs to use Q containing both Q_{MBEC} and Q_\theta since we want to (1) train f_{\Beta} and (2) use Q_{MBEC} as part of the target to train Q_\theta. (2) enables us to slowly integrate episodic values to the weight of Q_\theta, which models a consolidation process.
>
> $\textbf{"It seems more effective ..."}$ We view it differently. Purely estimating Q-value via minimizing normal TD-error is slow since, in the beginning, Q_\theta gives a poor estimation of the value. Adding Q_{MBEC} to the target provides better value estimation in the early phase of learning, which hastens the learning of Q_\theta. That compensates for the effort required to correct Q_{MBEC} (if any).
>
> $\textbf{"I don't fully understand the mechanics ..."}$ There are typos in the algorithm: L12-16 should be inside the "for" loop. We agree that the name "update" is a bit confusing and will change it to "boostrapped_write"
>
> $\textbf{"Is the primary benefit of ..."}$ The role of trajectorial loss is to help learn a good trajectory representation. If we have a good representation, the side effect will be both the ability to differentiate in partially observable environments and approximate the future. As stated in L117, we formulate the loss to reconstruct the $\textit{next}$ observation of $\textit{past}$ experience. It requires both capabilities: transition prediction and past reconstruction.
>
> $\textbf{Model-based baseline}$ We want to improve episodic control (EC) so we compare with SOTA EC, not model-based RL methods. Also, the Atari setting of Dreamer-v2 (200M frames) and SiMPLe (low-regime) is different from the common episodic control setting (10M), which makes it harder to compare. SiMPLe shows impressive results at 100K, yet still far from human-level performance. When trained with more frames, it is on par with model-free PPO. If we just look at the reported numbers, ours is far better than SiMPLe and if we check the learning curves of Dreamer-v2 at 10M frames, ours also shows competitive results. That comparison may not be fair, yet it gives a rough view of how our approach performs w.r.t recent model-based RL.
>
> $\textbf{Clarity}$ We appreciate your suggestion and will revise our paper based on your comments.
>
> $\textbf{L29 }$ You are right. We will add a point of partially observable environments. Thank you for your suggestion.
>
> $\textbf{L118}$ As stated in Appendix L508, we add Gaussian noise to the observations.
>
> $\textbf{L190}$ The feature vector is the output (before softmax layer) of a pretrained CNN (Resnet18). Its dimension is 512, easier to reconstruct than the raw image. This was mentioned in L200. We will add more details in the revised version.
>
> $\textbf{Table 1}$ It means the original papers do not report the numbers. EVA only reports learning curves (the results look inferior to ours anyway) and ERLAM does not run for 57 games.
>
> $\textbf{"The authors of the algorithms ..."}$, you are right about EVA. We will change "EEVA" to "EVA". However, ERLAM [1] is a different and stronger baseline than NEC2DQN [2].
>
> [1] Zhu, Guangxiang, Zichuan Lin, Guangwen Yang, and Chongjie Zhang. "Episodic reinforcement learning with associative memory." (2020).
>
> [2] Nishio, Daichi, and Satoshi Yamane. "Faster deep q-learning using neural episodic control." In 2018 IEEE 42nd Annual Computer Software and Applications Conference (COMPSAC), vol. 1, pp. 486-491. IEEE, 2018.

---

> > ### Comment · Reviewer_trSB · 2021-08-18
> > **Clarity and baselines**
> >
> > Thank you very much for your reply, and for clarifying some issues. I'm happy to see the points addressing the clarity of the model description and the algorithm.
> >
> > My primary remaining concerns are around (a) justifying the overall complexity of the model and (b) the comparison to model-based baselines, the latter of which was raised by other reviewers. I'm happy to see your response to reviewer BrPf on this point; a more robust comparison to model-based methods, with performance comparison, would put me more firmly on the accept side.
> >
> > I see that reviewer G6K7 suggested training the model-based baselines to 10M, which would be ideal. Another possibility is to meet halfway with the 100k regime and compare performance at 1M steps (this is already published for SimPLe in Appendix F of "Model-based Reinforcement Learning for Atari", though it's in graph form and normalized to baseline scores).

---

> > > ### Author Response · Authors · 2021-08-23
> > > **Baseline results**
> > >
> > > Thank you for your suggestion. We have quickly run the baseline and shared the initial results above. For SIMPLE, Appendix F does not report concrete numbers. To make the comparison clear, we reran our method on 100K steps setting and compared with SIMPLE's numbers on Table 2. We hope that the preliminary results give the reviewer more confidence to accept our paper.

---

> > > > ### Comment · Reviewer_trSB · 2021-08-30
> > > > **Re: Baseline results**
> > > >
> > > > Hello,
> > > >
> > > > I apologize for the late reply. I'm happy to see the new comparisons to model-based methods and that it is competitive with SOTA in this field. For this and the above clarity improvements, I'm raising my score to a 7.
> > > >
> > > > If accepted, any additional results you have against the model-based baselines by the final deadline would help to round out the analysis.

---

> > > > > ### Author Response · Authors · 2021-08-31
> > > > > **Thank you**
> > > > >
> > > > > Thank you for increasing your score. We will include these results in the revised manuscript.

---

### Author Response · Authors · 2021-08-10
**General reply**

Dear reviewers,

We appreciate your valuable feedback. Many of your comments are relevant and definitely will help us improve our paper. However, there remain misunderstandings that may hinder a proper evaluation. We will address these by replying to each of your reviews. We hope that our responses will address your concerns. Please consider increasing your score if you find our responses valid.

---

### Author Response · Authors · 2021-08-23
**More empirical evidence supporting our method**

Dear Reviewers,

We highly appreciate your constructive comments. We agree with the
reviewers that conducting a fair comparison with some model-based
baselines will demonstrate further the effectiveness of our method.
However, we would like to highlight that given our compute resources,
this is not an easy job to run additional baselines for all Atari
games. Hence, we decided to run the baselines on some games. We will do our best to perform more runs and will
add the final results to Appendix Fig. 9.

In particular, we have been running the baseline Dreamer-v2 for 10M
frames on 6 games. For now, we can share the preliminary result of 2 runs as
follows,

|  Model |  Alien |  Asterix | Breakout | Fishing Derby |  Gopher |  Hero |
|-----|----|---|---|----|---|---|
| Dreamer-v2 run 1 | **2950.1** | 3100.8 | 9.5 | -5.0 | **15888.4** | **13552.9** |
| Dreamer-v2 run 2 | 1830.4 | 2500.7 | 16.2 | -13.6 | 13960.7 | 11646.1 |
| Our MBEC++ | 1991.2 | **7724.0** | **372.2** | **17.6** | 11386.4 | 12148.5 |




For SIMPLE, we tried and realized that the deterministic SIMPLE--the
lightest among SIMPLE versions, ran much slower than our MBEC++ (10
steps/s vs 100 steps/s on our GPU, respectively). Hence, we could
not run the baseline for 10M frames to have a proper comparison. Instead,
we reran our MBEC++ (without any tuning) for 400K frames (~100K
steps in SIMPLE) 3 times and compare the mean/std against the SIMPLE (deterministic
version) on 6 games.

| Model | Alien | Asterix | Breakout | Fishing Derby | Gopher | Hero |
|-----|----|---|---|----|----|-----|
SIMPLE | **378.3 ± 85.5** | 668.0 ± 294.1 | 6.1 ± 2.8 | -94.5 ± 3.0 | **510.2 ± 158.4** | 621.5 ± 1281.3 |
Our MBEC++ | 323.3 ± 79.3 | **746.6 ± 31.18** | **10.2 ± 1.4** | **-84.6 ± 2.5** | 486 ± 64.0 | **2058.5 ± 1233.5** |



We hope that these initial results provide evidence that our method
is competitive against recent model-based baselines, confirming our
conjecture in the response to Reviewer BrPf. Notably, our trajectory
model is much simpler than the other methods (we only have TR and
reward losses and our network is the standard CNN of DQN for Atari
games). If we adopt sophisticated losses and network architectures
as in Dreamer and SIMPLE, our performance can even be better.

Finally, we hope that when making the final decision, the reviewers
consider other strong points of our paper:

- The novelty of our ideas (trajectory episodic memory, memory-based
planning and dynamic consolidation)
- Theoretical motivations for proposed components in our model (Appendix
A)
- Experiments on various domains (grid-world, classical control, Atari,
3d navigation)

---

> ### Comment · Reviewer_grEv · 2021-08-23
> **Clarification**
>
> Thank you for the additional baselines, they are much appreciated! I was just curious as to how this set of six games were chosen (e.g. as in prior work, for representativeness, at random, etc).

---

> > ### Author Response · Authors · 2021-08-25
> > **Game selection**
> >
> > Thank you for asking. There are criteria for selecting the games based on the performance of DQN and prior Episodic Control (EC). We looked at the published results (Table 3, Neural Episodic Control paper) and randomly selected games satisfying each criterion, as follows:
> > - Both DQN and prior EC perform poorly: Breakout, Fishing Derby
> > - DQN clearly underperforms prior EC: Alien, Hero
> > - DQN is roughly on par with prior EC: Asterix, Gopher.

---

### Decision · Program_Chairs · 2021-09-28

**Decision:**

Accept (Poster)

**Comment:**

This paper improves on recent works that leverage both episodic and habitual learning by combining state-action episodic memories with parametric value functions like Deep Q-Network.

The strengths of the paper include:  The idea is compelling from a theoretical/neuroscience standpoint (i.e., complementary learning systems); The proposed trajectory embedding and memory mechanisms are novel; The proposed dynamic weighting between memory-based and parametric values estimates is also novel; The performance is strong.  On the other hand, the reviewers have concerns regarding the complexity of the proposed approach.


**Consistency Experiment:**

NeurIPS has a long history of experimentation. In 2014, NeurIPS ran an experiment in which 10% of submissions were reviewed by two independent committees to quantify the randomness in the review process. This year, we repeated a variant of this experiment to see how the quality of the review process has changed over time.  This paper was part of the experiment and was therefore assigned to two committees (consisting of reviewers, an Area Chair, and a Senior Area Chair) that reached independent decisions.  If both committees made the same recommendation, this recommendation was followed. If a single committee recommended acceptance, the paper was accepted (with the exception of a few cases in which the other committee identified what we considered a fatal flaw, e.g., an error in a key result).

Both committees reached the same decision: **Accept (Poster)**

The other committee assigned to the paper recommended **Accept (Poster)**.  You can find the other set of reviews, along with any follow up discussion with the authors here:
https://openreview.net/forum?id=Z9Kpr38Kx_